# Advancing our understanding of genetic risk factors and potential personalized strategies for pelvic organ prolapse

Natàlia Pujol-Gualdo [1,2✉], Kristi Läll[1], Maarja Lepamets [1], Estonian Biobank Research Team*, Henna-Riikka Rossi[2], Riikka K. Arffman[2], Terhi T. Piltonen [2], Reedik Mägi [1,4] & Triin Laisk [1,4]

Pelvic organ prolapse is a common gynecological condition with limited understanding of its genetic background. In this work, we perform a genome-wide association meta-analysis comprising 28,086 cases and 546,291 controls from European ancestry. We identify 19 novel genome-wide significant loci, highlighting connective tissue, urogenital and cardiometabolic as likely affected systems. Here, we prioritize many genes of potential interest and assess shared genetic and phenotypic links. Additionally, we present the first polygenic risk score, which shows similar predictive ability (Harrell C-statistic (C-stat) 0.583, standard deviation (sd) = 0.007) as five established clinical risk factors combined (number of children, body mass index, ever smoked, constipation and asthma) (C-stat = 0.588, sd = 0.007) and demonstrates a substantial incremental value in combination with these (C-stat = 0.630, sd = 0.007). These findings improve our understanding of genetic factors underlying pelvic organ prolapse and provide a solid start evaluating polygenic risk scores as a potential tool to enhance individual risk prediction.

[1] Estonian Genome Centre, Institute of Genomics, University of Tartu, Tartu, Estonia. [2] Department of Obstetrics and Gynecology, PEDEGO Research Unit, Medical Research Centre, Oulu, University Hospital, University of Oulu, Oulu, Finland. [4] These authors contributed equally: Reedik Mägi, Triin Laisk. *A list of authors and their affiliations appears at the end of the paper. ✉email: natalia.pujol.gualdo@ut.ee

Pelvic organ prolapse (POP) is characterized by a descent of pelvic organs into the vaginal cavity[1]. It affects around 40% of women after menopause[2–4] and the lifetime risk of gynecological surgery for POP is up to 19% in the general female population[5]. The main symptoms include a bothersome sense of vaginal bulge, urinary, bowel, and/or sexual dysfunction, which substantially affects a woman's quality of life[6,7]. The most common risk factors are age, number of offspring, operative vaginal delivery, and BMI[8–10]. However, despite its health and economic impact, the etiology of this complex disorder remains poorly understood and there is a lack of evidence for early detection of women who are at risk of developing POP.

Genetic factors have been estimated to explain 43% of the variation for POP risk[11], yet have been poorly characterized. A recent genome-wide association study (GWAS) of Icelandic and UK Biobank (UKB) cohorts reported seven loci that associate with POP[12] and point to a role of connective tissue metabolism and estrogen exposure in its etiology. Nevertheless, increasing the sample size is likely to boost the power for the detection of more common risk variants, promoting the identification of genetic risk factors and enlightening biological mechanisms underlying POP.

Created based on GWAS results, polygenic risk scores (PRS) have been utilized as a tool to stratify individuals into different risk groups in complex diseases[13]. However, PRS have never been used as a tool for predicting POP development. In contrast to physical examinations, which remain to be the gold-standard for POP assessment, PRS could serve as a tool to identify women with higher genetic risk of POP and tailor individualized preventive strategies even decades before symptomatic POP appears.

In this work, we present the largest GWAS for POP to date (nearly doubling the number of cases compared to previous efforts[12]) and systematically dissect the association signals to propose potential causal genes in associated loci. We also developed for the first time a PRS for POP to evaluate risk stratification using individual genetic risk and the predictive ability of PRS alone or in combination with classical risk factors, a tool that might favor preventive and personalized strategies in the future.

## Results

**Genome-wide inferences.** We performed a meta-analysis with data from three studies (Icelandic and UKB cohorts, FinnGen study, and EstBB) including a total of 28,086 women with POP and 546,291 female controls of European ancestry (Supplementary Fig. 1). The meta-analysis identified a total of 26 loci, with 30 independent lead signals significantly associated with POP ($P < 5 \times 10^{-8}$) (Supplementary Data 1). From these, 19 loci were novel findings and we replicated all seven previously reported loci[12]. According to $I^2$, two variants showed large heterogeneity (rs3820282 $I^2 = 76\%$ and rs72624976, $I^2 = 83\%$, the latter likely due to differences in allele frequency between cohorts (Supplementary Data 2), although Q-Cochran test showed no heterogeneity of effects between the three datasets at any of the lead signals (Q-Cochran $p$ varying from 0.0023–0.982, with a Bonferroni corrected threshold of 0.05/30 = 0.001) (Supplementary Fig. 2, Supplementary Data 1). All lead variants were present in at least two out of the three datasets analyzed and were common variants (MAF > 0.05) except for one replicated (rs72624976, EAF = 0.01, $p = 1.14 \times 10^{-9}$) and one previously unidentified association (rs72839768, EAF = 0.02, $p = 4.66 \times 10^{-9}$). There was no evidence of excessive genomic inflation ($\lambda = 1.054$) in the GWAS meta-analysis (LDSC intercept = 1.0059 (s.e. 0.0079)) (Fig. 1). The observed SNP heritability estimate was 0.017 (s.e. 0.001), which corresponds to a liability scale SNP heritability of 0.079 given a population prevalence of 0.05 for symptomatic POP and 0.143 given a population prevalence of 0.40 for overall POP.

The SNP heritability explained by the significant loci identified was 0.022 given a population prevalence of 0.05.

**Gene prioritization.** In order to move from genetic variants to plausible candidate genes, we prioritized genes according to different data layers of evidence, considering at least the presence from one of the next four main evidence levels: (1) positional mapping as implemented in FUMAv1.3.6a[14] was used to determine the nearest gene to the association peak; (2) genes containing variants which showed significant (posterior probability >0.8) colocalization in eQTL datasets; (3) genes containing non-synonymous variants or in high LD (r2 > 0.6) with these; (4) genes which showed embryo, growth/size/body, muscle, renal/urinary system, reproductive system, digestive/alimentary system phenotypes in mutant mice (Mouse Genome Informatics[15], MGI).

Overall, our results are in line with previous findings, replicating and highlighting associations near *WNT4*, *EFEMP1*, *FAT4*, *IMPDH1*, *TBX*, and *SALL1*[12]. Colocalization analyses allowed us to highlight an additional candidate gene, *LDAH* in the previously reported 2p16.1 locus. Additionally, significant colocalization signals allowed us to prioritize potential previously unidentified candidate genes such as *VCL*, *CHRDL2*, *DUSP16*, *LOXL1-AS1*, *CRISPLD2*, *KLF13*, *ADAMTS5*, and *MAFF* in 2p24.1, 10q22.1, 11q13.4, 12p13.2, 15q24.1, 15q13, 16q24.1, 21q21.3, and 22q13 respectively (Fig. 2 and Supplementary Data 3). Data from mouse models also supported the roles of previously unidentified candidate genes such as *ACADVL* (*Acadvltm1Vje/Acadvltm1Vje*), *PLA2G6* (*Pla2g6m1Sein/Pla2g6m1Sein*), and *HOXD13* (Hoxd13tm1Ddu/Hoxd13+), which have been associated with muscle fiber formation, muscle weakness and muscle hypotonia phenotypes. *Wt1*<sup>tm2Asc</sup>/*Wt1*<sup>tm2Asc</sup> mouse models exhibited abnormal reproductive physiology and *GREM1* mouse models exhibited diverse renal and urinary system abnormalities. Additionally, we confirmed previously reported candidate genes with substantial evidence from mice model studies such as *LOXL1* and *EFEMP1*; a mouse model knock-out for *LOXL1* (*Loxl1tm1Tili/Loxl1tm1Tili*) exhibited uterus prolapse and dilated uterine cervix[16], while *EFEMP1* knock-outs exhibited decreased skeletal muscle weight, loose skin, and abnormal urogenital development.

Based on functional impact, we were able to identify three additional candidate genes for POP. The lead variant in 17p13.1 (rs72839768, $p = 4.66 \times 10^{-9}$) is a missense variant of the *DVL2* gene. Additionally, two non-synonymous variants in LD with the lead signals were identified in two genes, *LOH12CR1* in 12p13 (rs3751262, $p = 2.89 \times 10^{-7}$, $r^2 = 0.70$) and *LACTB2-AS1* in 8q13.2 (rs35863913, $p = 3.05 \times 10^{-7}$, r2 = 0.73).

**Gene set and tissue/cell-type enrichment.** Gene set and tissue/cell-type enrichment analysis implemented in MAGMA[17] and DEPICT[18] highlighted "Connective Tissue Development" ($p = 2.01 \times 10^{-6}$), "Chondrocyte differentiation" ($p = 6.63 \times 10^{-4}$) and "In utero embryonic development" ($p = 4.91 \times 10^{-8}$), Supplementary Fig. 3 and Supplementary Data 4 and 5). 12 tissues were significantly enriched after correcting for multiple testing, including "Cervix/ectocervix" ($p = 1.30 \times 10^{-5}$), "Uterus" ($p = 1.50 \times 10^{-5}$), "Embryoid bodies" ($p = 8.60 \times 10^{-6}$) and "Smooth muscle" ($p = 7.30 \times 10^{-4}$; Supplementary Fig. 3 and Supplementary Data 6 and 7).

**Genetic correlation.** Genetic correlation with POP was estimated through pairwise comparison with published GWASs (43 phenotypes) and GWASs of UK Biobank data (518 phenotypes), totaling 561 phenotypes, using LD score regression implemented in LD-Hub[19,20]. 90 phenotypes demonstrated significant genetic overlap with POP ($p < 8.91 \times 10^{-5}$) (Fig. 3 and Supplementary Data 8). We observed the largest positive correlation with hysterectomy ($r_g = 0.59$, $p = 3.43 \times 10^{-17}$). POP was positively correlated with the number of children ($r_g = 0.22$, $p = 2.82 \times 10^{-8}$), whilst age at first live birth was negatively correlated ($r_g = -0.19$,

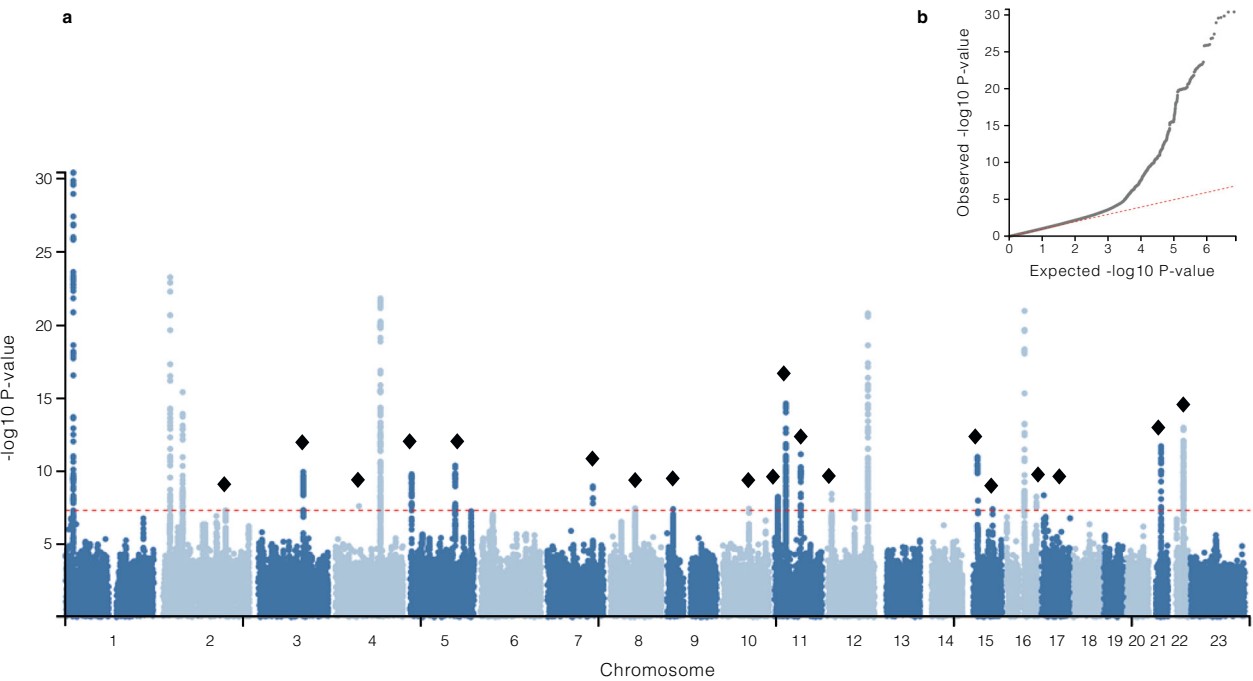

**Fig. 1 Manhattan plot showing genome-wide significant loci associated with pelvic organ prolapse and QQplot. a** Manhattan plot for GWAS meta-analysis for pelvic organ prolapse The novel candidates are highlighted as a black diamond. The $y$ axis represents $-\log_{10}(P\text{-values})$ for association of variants with POP, from mixed logistic regression analysis of cohorts (adjusted by year of birth and 10 principal components). The horizontal dashed line represents the threshold for genome-wide significance ($P < 5 \times 10^{-8}$). **b** QQ plot. The panel displays a QQ plot, which show the $-\log_{10}(P\text{-values})$ based on a two-sided $Z$-tests for all SNPs. The dotted line represents the expected $-\log_{10}(P\text{-values})$ under the null hypothesis.

$p = 1.42 \times 10^{-7}$). Positive associations were observed with gastroesophageal reflux ($r_g = 0.31$, $p = 5.37 \times 10^{-7}$), diverticular disease ($r_g = 0.28$, $p = 4.33 \times 10^{-16}$), osteoarthritis ($r_g = 0.23$, $p = 4.48 \times 10^{-6}$), hiatus hernia ($r_g = 0.32$, $p = 6.68 \times 10^{-5}$) and abdominal and pelvic pain ($r_g = 0.31$, $p = 3.58 \times 10^{-7}$). We additionally saw positive correlations with traits such as excessive frequent and irregular menstruation ($r_g = 0.36$, $p = 3.47 \times 10^{-5}$) and several cardiovascular phenotypes: coronary artery disease ($r_g = 0.16$, $p = 4.41 \times 10^{-5}$), angina ($r_g = 0.19$, $p = 8.79 \times 10^{-5}$) and myocardial infarction ($r_g = 0.22$, $p = 3.05 \times 10^{-5}$). Positive correlations were also observed for traits reflecting type of occupation: job involving mainly walking or standing ($r_g = 0.19$, $p = 3.7 \times 10^{-7}$) and job involving heavy manual or physical work ($r_g = 0.21$, $p = 4.4 \times 10^{-9}$). Genetic correlations related to obesity include significant relationships with body mass index (BMI) ($r_g = 0.12$, $p = 4.73 \times 10^{-7}$), waist-to-hip ratio ($r_g = 0.20$, $p = 4.27 \times 10^{-8}$), triglycerides ($r_g = 0.17$, $p = 1.6 \times 10^{-6}$) and diabetes diagnosed by doctor ($r_g = 0.15$, $p = 3.64 \times 10^{-5}$).

Phenome-wide associations. We performed a phenome-wide association look-up of associated variants using GWAS Catalog[21] and Phenoscanner V2 databases[22], which underlined several traits spanning abnormality of connective tissue, body measurements, cancer, cardiovascular disease, digestive system disorders, pulmonary function, reproductive health, liver disease, psychiatric disorders and other traits (Supplementary Figs. 4, 5 and Supplementary Data 9, 10).

Developing PRS for POP. We compared the following two software for calculating PRS: LDPred1.0.11[23] and PRSice[24]. The resulting PRSs associations were scaled as follows: we calculated odds ratio (OR) per one standard deviation of the PRS and 95% confidence intervals (95% CI). We found that the best performing PRS consisted of 3,242,959 SNPs and was built by LDPred. This version showed an OR = 1.42 (95% CI 1.37 to 1.47) and $p = 2.59 \times 10^{-89}$ towards the case-control discrimination in the

discovery set (including 5379 prevalent cases and 21,516 controls) (Supplementary Fig. 6 and Supplementary Data 11).

PRS performance across categories. We proceeded to test the predictive ability of the PRS in the validation set (totaling 2517 incident cases and 96,109 controls). Analyzing the best PRS in the validation set, a continuous PRS distribution showed the highest Harrell's C-statistic (C-stat) of 0.616 (sd = 0.006). In the validation set, we observed a risk gradient within percentiles (Fig. 4). Women in the top 5% of the PRS distribution had 1.61 (95% CI: 1.35–1.92) times the hazard of developing POP compared to the rest of the women and 1.53 (95% CI: 1.26–1.86) times the hazard compared to women from the average (40–60%) (Supplementary Data 13). However, it is important to note that this is a Kaplan–Meier estimate, which does not take into account competing risks such as death before developing the condition, and thus incidence rates might be overestimated. When assessing genetic risk in different age categories, none of the age groups showed a higher HR compared to the full validation set analysis (Supplementary Data 13). Women in 50–60 years old strata showed highest HR amongst categories when comparing top 5% genetic risk vs rest of women (HR = 2.04, 95% CI = 1.42–2.70), as well as highest number of incident cases in top 5% genetic risk ($n = 834$) compared to younger and older strata (Supplementary Data 12).

Predictive ability of the PRS and clinical variables. From the validation set of EstBB we further selected a validation subset of 2104 cases and 24,753 controls who had little or almost no missing clinical covariate data (Supplementary Fig. 1, Supplementary Data 14), which allowed us to test the predictive ability of the PRS alone or in combination with 5 clinical variables (number of children, BMI, ever smoked, constipation and asthma). In the validation subset, the continuous PRS distribution showed a C-stat of 0.583 (sd = 0.007). Amongst the clinical risk factors evaluated, number of children was the best predictor

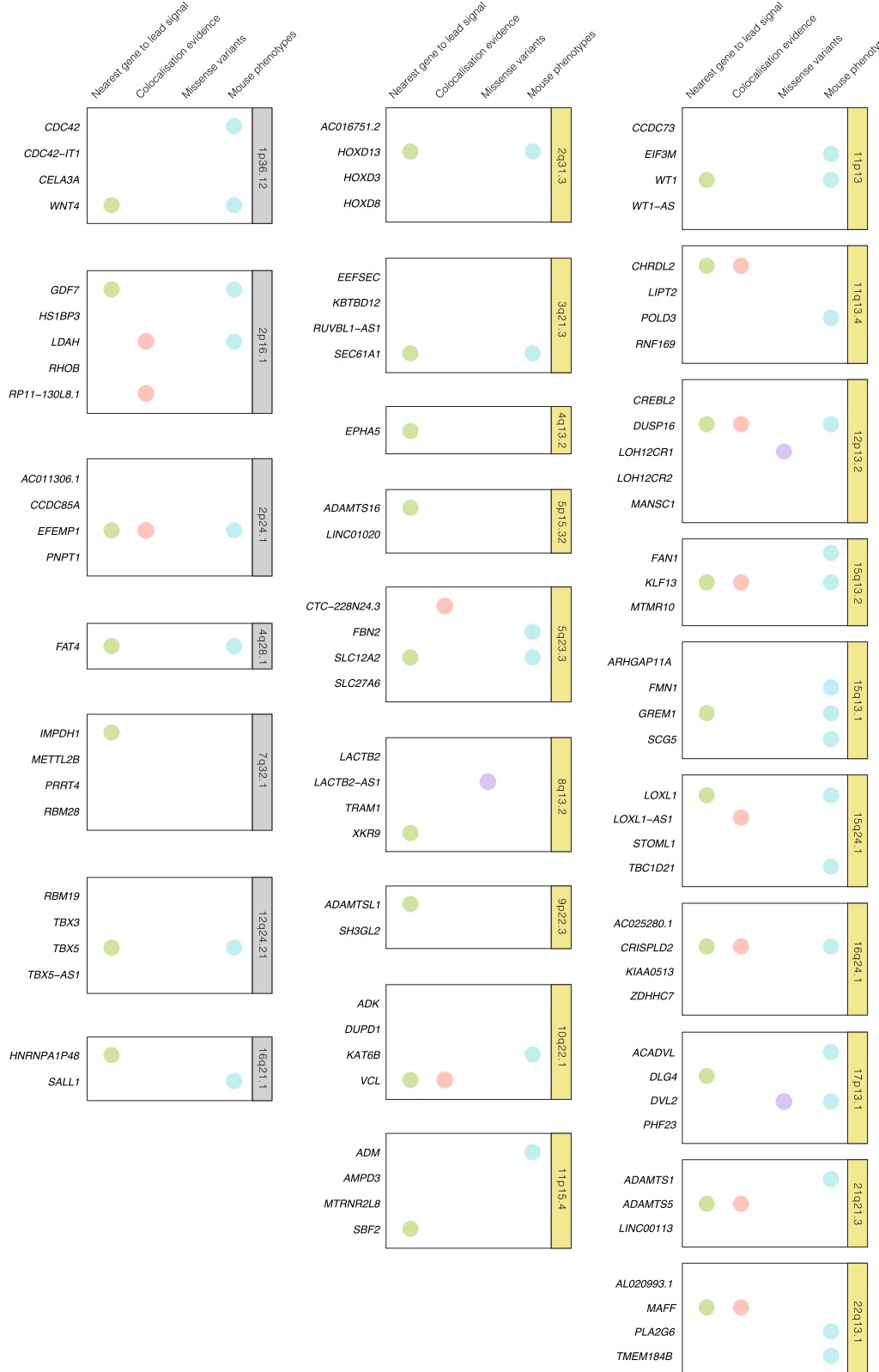

**Fig. 2 Evidence for pelvic organ prolapse GWAS meta-analysis candidate gene mapping.** Previous reported loci are highlighted in gray frames and previously unidentified loci in yellow frames. We prioritized candidate genes considering at least the presence of one of the next four main evidence levels: (1) nearest gene to the association peak (indicated by green dots); (2) genes containing shared causal variants between genetic variants and gene expression signatures unraveled by colocalization analyses (shown as orange dots); (3) genes containing coding variants or in high LD (r2 > 0.6) with these (shown as purple dots); and (4) genes which showed embryo, growth/size/body, muscle, renal/urinary system, reproductive system, digestive/alimentary system phenotypes in mutant mice.

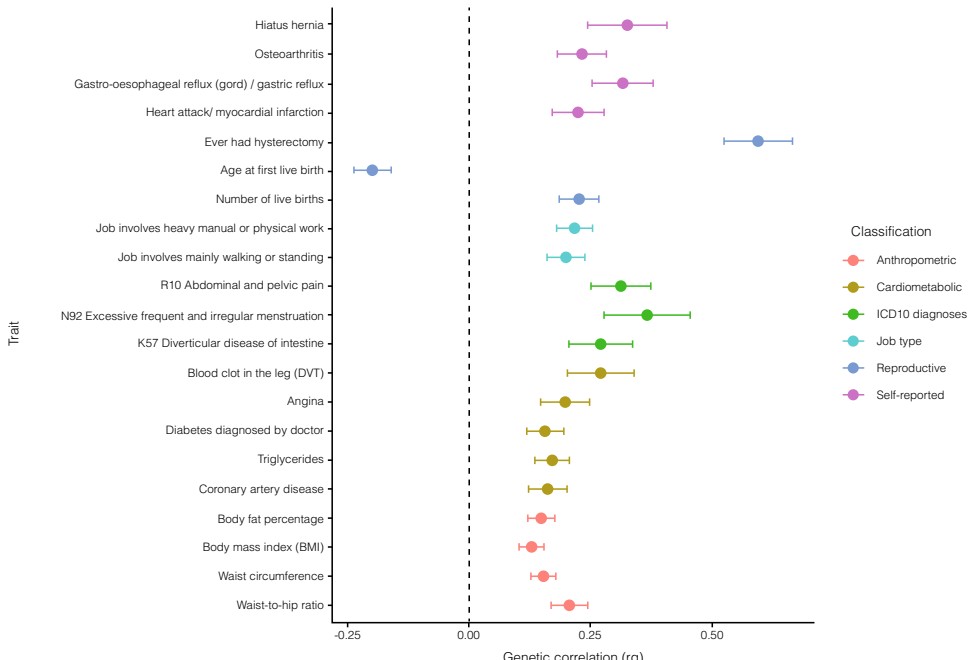

**Fig. 3 Genetic correlation analyses.** The genome-wide genetic correlation of POP GWAS meta-analysis summary statistics with 561 phenotypes was estimated using LDSC regression. Data is presented as means +− SEM. We accounted for multiple testing using a Bonferroni correction for 561 tests $(0.05/561 = 8.91 \times 10^{-5})$ and derived genetic correlation estimates (showed as circles). Phenotypes summary statistics come from published GWASs $(n = 43$ phenotypes) and GWASs of UK Biobank data $(n = 518$ phenotypes), available in LD-Hub v1.9.3. Significant genetic correlations showcased in the plot reveal overlap of genetic risk factors for POP across several groups of traits (grouped by colors): anthropometric (red dots; including body fat percentage $(n = 354,628)$, body mass index $(n = 354,831)$, waist circumference $(n = 360,564)$, waist-to-hip ratio $(n = 224,459)$, cardiometabolic (yellow dots; including blood clot in the leg (DVT) $(n = 7,386$ cases and 353,141 controls), angina $(n = 11,372$ cases and 349,048 controls), diabetes diagnosed by doctor $(n = 17,275$ cases and 342,917 controls), triglycerides (mmol/L) $(n = 343,992)$, coronary artery disease $(n = 60,801$ cases and 123,504 controls)), ICD10 diagnoses (green dots; including R10 Abdominal and pelvic pain $(n = 20,240$ cases and 340,954 controls), N92 Excessive frequent and irregular menstruation $(n = 8,475$ cases and 185,699 controls), K57 Diverticular disease of intestine $(n = 12,662$ cases and 348,532 controls), job type (light blue dots; including Job involves heavy manual or physical work $(n = 205,000)$, Job involves mainly walking or standing, $n = 204,956)$, reproductive traits (dark blue dots; including Ever had hysterectomy $(n = 13,973$ cases and 157,440 controls), Age at first live birth $(n = 131,987)$, Number of liver births $(n = 193,953))$ and self-reported conditions (pink dots; including Hiatus hernia $(n = 32,590)$, Osteoarthritis $(n = 30,046$ cases and 331,095 controls), Gastro-esophageal reflux $(n = 15,210$ cases and 345,931 controls), Heart attack/myocardial infarction $(n = 8,239$ cases and 352,902 controls)). Study source can be found in Supplementary Data 8. Center values show the estimated genetic correlation (rg), which is presented as a dot and error bars indicate 95% confidence limits. Dotted black line indicates no genetic correlation. ICD International Classifications of Diseases 10th Revision, DVT Deep Venous Thrombosis.

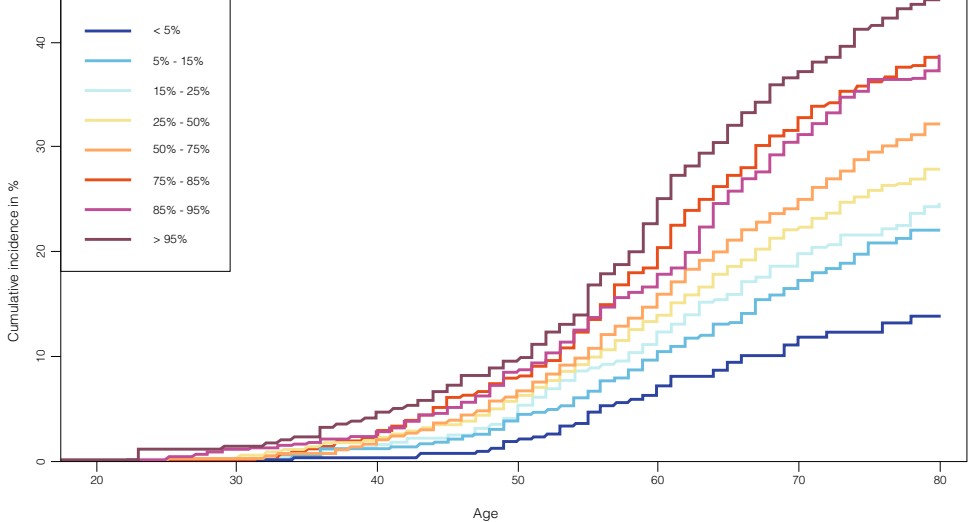

**Fig. 4 Cumulative incidence by PRS categories.** Cumulative incidence of POP in % scaled by age in the validation set of Estonian Biobank (2517 incident cases and 96,109 controls) for different POP PRS percentiles (<5%, 5–15%, 15–25%, 25–50%, 50–75%, 75–85%, 85–95%, >95%). Survival modeling and Cox proportional hazard models were implemented, using age as a time scale for properly accounting for left-truncation and right-censoring in the data.

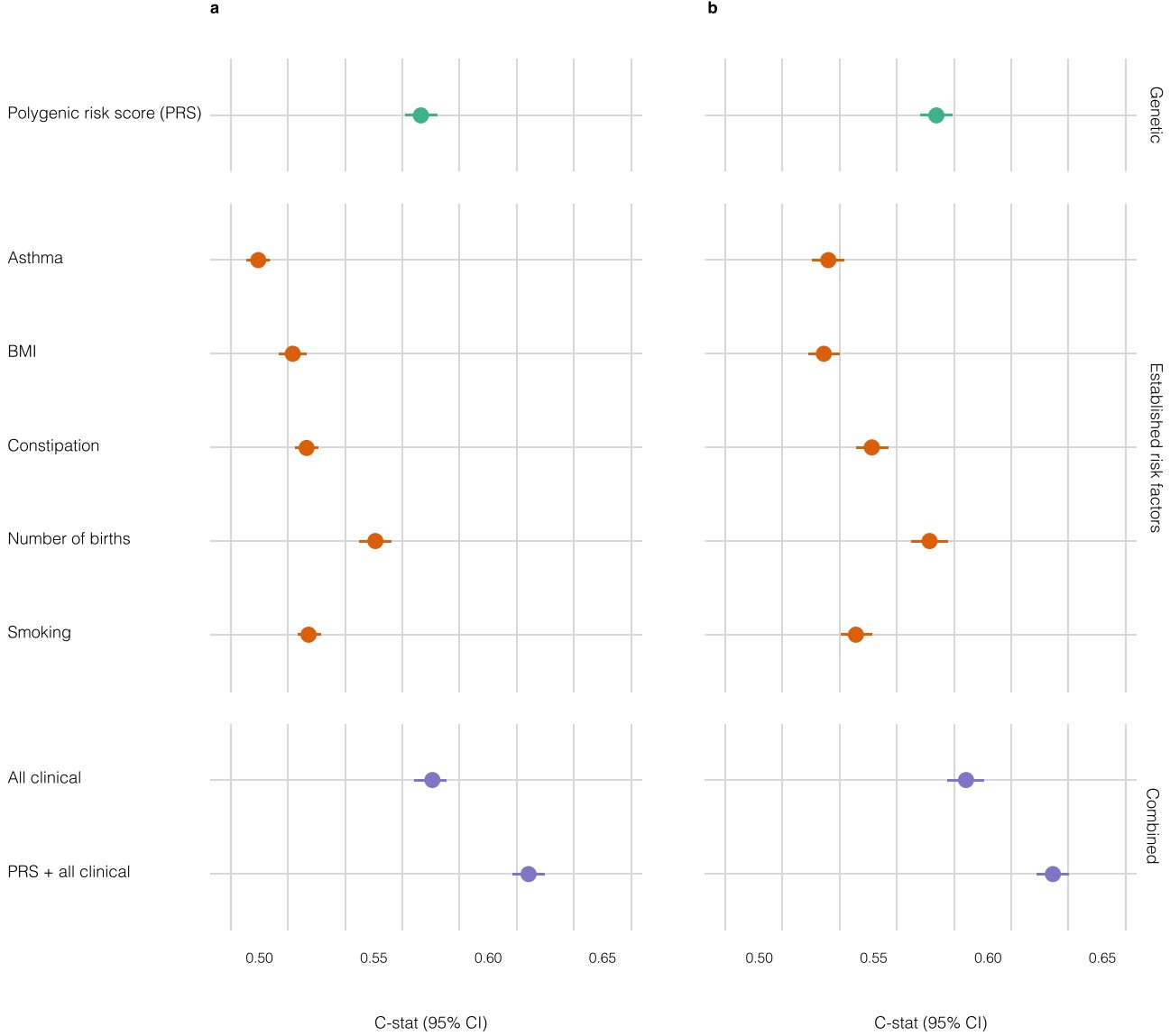

**Fig. 5 Predictive ability of PRS and clinical variables in incident status.** Green dots represent polygenic risk score (PRS), orange dots represent five established risk factors and purple dots represent genetic and/or clinical combined models C-statistic (C-stat) indexes. Data are presented as means +/− SEM in both panels. Cox proportional hazard models were used and age was used as a time scale for properly accounting for left-truncation and right-censoring in the data in both models. **a** C-stat for clinical variables and PRS alone or in combination in the validation subset of Estonian Biobank (2104 cases and 24,753 controls). **b** C-stat adjusted by batch effects and 10 first principal components in the validation subset of Estonian Biobank (2104 cases and 24,753 controls).

(C-stat 0.563, sd = 0.006), followed by ever smoked (0.534, sd = 0.005), constipation (0.533, sd = 0.005), BMI (0.528, sd = 0.006) and asthma (0.512, sd = 0.005). Adding PRS in the clinical joint model notably improved the predictive ability (0.630, sd = 0.007) compared to only the clinical joint model containing the five clinical risk factors (0.588, sd = 0.007) (see Fig. 5 and Supplementary Data 15).

## Discussion
This study constitutes the largest GWAS meta-analysis for POP, uncovering 26 genetic loci (19 novel), which represents almost a four-fold increase in the number of loci associated with POP compared with the previous study[12]. In contrast to previous efforts characterizing genome-wide signals underlying POP[12], we use a combination of different data layers to map potential candidate genes, which provide novel insights into the biology of prolapse development, open up new avenues for further

functional studies, and identify links with other health outcomes, which might have important implications for patient management and counseling. Additionally, we construct for the first time a PRS for POP, which shows similar or better predictive ability than five risk factors (number of children, constipation, BMI, asthma, and smoking status), and we demonstrate that PRS in combination with these clinical risk factors generates the best predictive model for POP.

Among the genome-wide significant loci, this study supports the role of a previous reported candidate gene for POP, *LOXL1*. Liu et al. described that mice lacking the protein lysyl oxidase-like 1 (LOXL1) do not deposit normal elastic fibers in the uterine tract postpartum and develop POP[16]. Subsequently, diverse mouse and human studies have reiterated its involvement with prolapse[25–27]. Additionally, we further support previous findings highlighting associations near *WNT4*, *EFEMP1*, *FAT4*, *IMPDH1*, *TBX5*, and *SALL1*. Beyond *EFEMP1*, we further propose several previously

unidentified candidate genes (*CHRDL2, ACADVL, PLA2G6*) which reinforce the role of connective tissue molecular changes as a key process in the pathogenesis of POP[28,29]. These links are mirrored by positive genetic correlation between POP and several connective tissue abnormalities such as inguinal hernia. Additionally, our study highlights a plausible role of A Disintegrin And Metalloprotease with Thrombospondin Motifs (ADAMTS) in POP, which have important roles in extracellular matrix maintenance[29]: on the locus described in chromosome 21, the most plausible candidate gene was *ADAMTS5*. A recent study described genetic associations between ADAMTS/ADAMTSL members and inguinal hernia[30], which also showed significant associations (5p13.2/*ADAMTS16* and 9p22.3/*ADAMTSL1*) in our study.

Our study also reinforces urogenital development as a key process in the pathogenesis of POP highlighting previously unidentified genes such as *DVL2, WT1, HOXD13*. *DVL2* is a component of the Wnt signaling pathway (a shared pathway with the previously reported gene *WNT4*), important for epithelial tissue development and renewal, embryonic development of the sex organs, and regulation of follicle maturation controlling steroidogenesis in the postnatal ovary[31–34]. Several in vitro and animal studies have shown that estrogen has positive effects on the ECM[35,36] and a study described a link between hypoestrogenism and deterioration of the ECM and concomitant POP[37], indicating the Wnt pathway may have a dual role in POP development, both by regulating organogenesis and hormonal support of tissue function. Future studies assessing sex-specific oestradiol exposure with larger sample sizes and more scaled time points, might inform the direction of association between estrogen lifetime exposure and the development of POP.

Additionally, *WT1*, another proposed novel candidate gene, is a transcription factor involved in urogenital system development. Recently, a single-cell transcriptome profiling study of severe anterior vaginal prolapse described activity of WT1 in fibroblasts, and genes regulated by WT1 were enriched in terms related to actin filament behavior[38]. Moreover, *WT1* has also been involved in cardiac development and disease[39–41], a link that was further supported by our phenome-wide association look-up results, since this region was associated with hypertension and cardiovascular disease (Supplementary Figs. 4, 5 and Supplementary Data 9 and 10). In this line, many GWAS associations we report point towards a link between metabolic and cardiovascular health and POP (*KLF13, DUSP16, MAFF, VCL*, and *LDAH*), which is a particularly interesting genetic association unraveled by our study. These associations were mirrored by positive genetic correlations with a range of cardiometabolic phenotypes[42–51].

Genetic correlation analyses also highlighted a positive correlation with occupations involving walking and/or standing and heavy physical work, which might have potential value in counseling of women with higher risk to develop POP. While unfortunately, it is not possible to discern the timing of hysterectomy in relation to POP in the present study, future linkages and/or cohorts with larger availability of POP-related procedures might better inform the effect of hysterectomy as both cause and consequence to POP.

Genetic correlation studies mirror well the findings of epidemiological studies, showing associations with number of births, previous hysterectomy, younger age at first birth, increasing BMI, constipation, occupations including heavy lifting and connective tissue disorders[8,52,53].

The association between POP and gastroesophageal reflux, diverticular disease, osteoarthritis, and hiatus hernia most likely reflects the changes to connective tissue characteristic to all of these conditions; however, respective epidemiological links have been inconsistent. Similarly, future epidemiological studies assessing the association with abdominal and pelvic pain,

excessive frequent and irregular menstruation, and cardiometabolic phenotypes are warranted.

Here we present a comprehensive GWAS follow-up which culminated in defining possible convergence in meaningful biological pathways through defined test sets of genes such as "Connective Tissue Development" ($p = 2.01 \times 10^{-6}$), "Chondrocyte differentiation" ($p = 6.63 \times 10^{-4}$), and "In utero embryonic development" ($p = 4.91 \times 10^{-8}$) and link these to specific tissues and cell types such as "Cervix/ectocervix" ($p = 1.3 \times 10^{-5}$), "Uterus" ($p = 1.5 \times 10^{-5}$), "Embryoid bodies" ($p = 8.6 \times 10^{-6}$) and "Smooth muscle" ($p = 7.3 \times 10^{-4}$). These findings might aid in choosing the relevant tissue or cell type for in vitro experiments to further elucidate molecular mechanisms underlying the genome-wide significant loci identified.

The added value of our study in comparison to previous efforts[12] is best highlighted by presenting the first polygenic risk score for POP, which substantially advances the concept of using genomic information to stratify women in gynecological conditions. However, it is prudent to consider that common SNPs explain a small part of the whole heritability and it is plausible that most of SNP-heritability is yet to be discovered, which hinders assessing the full potential of PRS. Potential sources of missing heritability might include much larger numbers of smaller effect variants yet to be found; rarer variants (possibly with larger effects) that are poorly detected by available genotyping arrays; structural variants poorly captured by existing arrays, low power to detect gene-gene interactions, etc.[54]. Further studies with larger sample sizes are needed, which will enable comprehensive PRS performance comparisons and will improve the evaluation of genetic risk assessment.

In our study, predictive ability analyses in the validation set properly took into account the effect of age by including it as the time scale in the survival model, thus accounting for left-truncation and right-censoring in the data and comparing only women from same ages and avoiding inflation in prediction solely due to age difference between cases and controls.

The addition of the PRS in the clinical model clearly demonstrates a superior predictive ability in incident POP (C-stat=0.630) than when analyzing the five clinical risk factors without PRS (C-stat=0.588). It is also important to note that this type of huge improvement provided by a polygenic risk score on top of classical risk factors (+4.2 percentage points) is not a common finding—for example, a groundbreaking work by Inouye and colleagues[55] in coronary artery disease (CAD) revealed that six conventional risk factors achieved a C-stat of 0.67, and together with PRS it increased by 2.6 percentage points to 0.696, whereas PRS alone was a better predictor (C-stat=0.623) than any of six conventional factors (smoking, diabetes, hypertension, BMI, self-reported high cholesterol, and family history) alone—which is also the case for POP.

Contrary to a recently developed screening tool for pelvic floor disorders after delivery (http://riskcalc.org/UR_CHOICE/)[56], genetic risk is stable, and thus evaluable throughout the lifespan. Similar to other works for complex diseases, PRS might offer potential clinical uses in settings related to disease risk stratification and the encouragement to direct earlier preventative strategies (such as weight reduction, preventing constipation, Kegel exercises for pelvic floor muscle strengthening, and avoiding heavy lifting) to those women with higher genetic risk to develop POP, although the clinical translation of PRS profiling for early diagnosis and targeted screening needs to be further examined in future cohorts with longer follow-up time and an increased number of incident POP cases. Similarly, future studies assessing risk prediction towards those cases who present more severe forms—e.g., requiring surgical intervention- might open up new avenues for targeting clinical resources, increasing check-

up frequency and direct targeted preventive exercises and counseling for those women.

Phenotype definition variability can be critical to the successful identification of genetic associations. We observed that heterogeneity estimates of lead signals (taking into account Q-Cochran $p$-values and $I^2$) mostly do not substantially differ between studies, which on one hand adds reliability to the identified genetic associations and on the other hand supports leveraging electronic health records to ascertain this phenotype. According to $I^2$, two variants showed heterogeneity (rs3820282 $I^2 = 76,45\%$ and rs72624976, $I^2 = 83.46\%$). rs72624976 heterogeneity is likely due to differences in allele frequency between cohorts (see Supplementary Data 2 and Supplementary Fig. 1, with lower frequency for Finn-Gen and EstBB (EAF = 0.01) than Icelandic and UKBB (EAF = 0.04)). This observation emphasizes the importance of the existence of population-specific biobanks when studying genetics of complex diseases. rs3820202 heterogeneity might be explained by the presence of more severe cases in FinnGen and IceUK cohorts (perhaps due to age distribution differences between cohorts or source of diagnoses: hospital inpatient registry data for UKBB/ FinnGen which might capture more severe cases compared to EstBB, where diagnoses made by general practitioners or gynecologists can also include milder diagnoses). However, heterogeneity estimates tend to be imprecise when assessing heterogeneity in meta-analysis containing a small number of studies[57–59].

In this regard, one of the strengths of the study is the comprehensive data availability in EstBB, containing genetic data of around 20% of Estonian adult population including phenotype questionnaire and measurement panel, together with follow-up data from linkage with national health-related registries, which facilitated the validation of PRS and the inclusion of clinical risk factors into a joint model. In a similar way, the coverage and accuracy of the Finnish Care Register for Health Care has been validated previously[60,61] and it has been found to be excellent. Previous effort meta-analyzing UKB and Icelandic studies assessed the robustness of ICD codes by comparing effect sizes with surgically treated POP cases and concluded that effects were in the same direction and not substantially different from those using surgically treated POP cases[12].

Possible phenotypic misclassification (undiagnosed cases amongst controls) could result in heterogeneity in the analysis and interpretation of GWAS findings, reducing both the statistical power as well as the maximum number of significant associations observed their effects magnitude and direction.

We hope that future efforts replicate our findings, either independent or stemming from increased numbers of cases in new data linkages in the included datasets, and more extensive questionnaire data collection. Similarly, future questionnaires in the biobank setting could address different disease stages and severity. The present study is limited by the unavailability of other classical risk factors that would improve the predictive ability of the combined model, such as newborn anthropometric measurements, mode of delivery, or more extensive reproductive history. Since both GWAS meta-analysis and PRS construction were based on European ancestry populations, this challenges the generalizability of GWAS findings to other populations and warrants caution when extrapolating the results.

Future larger sample size in gene expression panels of relevant tissues in gynecological conditions is crucial, since it might add greater power to detect more significant associations and improve disentangling tissue-specific signals in eQTL datasets[62]. While recent evidence suggests distance to the association peak as a good predictor for a causal gene[63,64], it is important to note that complex LD patterns between association signals might eclipse more distant genes which are the true causal ones. Overall, further functional follow-up is needed to better characterize the regulatory functions of the loci uncovered and experimental work is warranted to unravel the role of the nominated genes.

In summary, this study is the first to explore PRS as a tool to inform risk stratification strategies for POP and highlights the potential to improve predictive ability when adding genetic risk on top of clinical risk factors. Additionally, we presented the most comprehensive analysis of genetic risk factors of POP to date, providing many candidate genes and defining possible convergence in meaningful biological pathways, which establishes a landmark step forward in the field of female reproductive genetics, moving from genetic signatures to a source of biological insight for POP.

## Methods

This study was carried out under ethical approval 1.1-12/624 from the Estonian Committee on Bioethics and Human Research (Estonian Ministry of Social Affairs) and data release N05 from the EstBB. All necessary patient/participant consent has been obtained and the appropriate institutional forms have been archived.

**Study cohorts**. Our analyses included a total of 28,086 women with POP and 546,291 controls of European ancestry from three different studies: summary level statistics from an Icelandic and UKBB GWAS meta-analysis[12] (IceUK, 15,010 cases, and 340,734 female controls), and FinnGen R3 (5518 cases and 43,366 controls), and individual level data from the Estonian Biobank (EstBB, 7896 cases, and 118,865 controls) (Supplementary Fig. 1). Cases were defined as women having POP diagnosed by ICD-10: N81, ICD-9: 618 and ICD-8: 623 upon availability. Controls were defined as individuals who did not have the respective ICD codes.

**Cohort and genotyping details**. Summary statistics from Icelandic and UKB cohorts meta-analysis were downloaded upon request, which included prefiltered variants selected based on a threshold of 0.8 imputation info and MAF > 0.01%, available in the Icelandic data set and/or the UKB dataset. Cohort and genotyping details from Icelandic and UKBB cohorts have been reported elsewhere[12].

FinnGen is a public-private partnership project combining data from Finnish biobanks and electronic health records from different registries. In this study, we used the results from the FinnGen release R3 (https://www.finngen.fi/en/access_results), which includes data from 135,638 individuals and more than 1800 disease endpoints and it is publicly available for download. FinnGen individuals have been genotyped with Illumina and Affymetrix arrays and imputed to the population-specific SISu v3 importation reference panel. Genetic association testing has been carried out with SAIGE. We downloaded the summary statistics querying the disease endpoint "Female genital prolapse", which included prefiltered variants (minimum allele count >5 and INFO score >0.6). 5518 individuals with the ICD10 N81 diagnosis were defined as cases and 43,366 controls were included in the analysis.

The Estonian Biobank (EstBB) is a population-based biobank with over 200,000 participants, currently including around 135,000 women (20% of Estonian female population). The 200 K data freeze was used for the analyses described in this paper. All biobank participants have signed a broad informed consent form. Individuals with POP were identified using the ICD-10 code N81 (mean age=58.76, sd=12.01), and all female biobank participants who did not have this diagnosis were considered as controls (mean age=43.86, sd=16.06), which included 7968 cases and 118,895 controls. Information on ICD codes is obtained via regular linking with the National Health Insurance Fund and other relevant databases[65]. We excluded from all analyses controls who did not have health registry information linked, since those might have not had the opportunity to register diagnose status.

All EstBB participants were genotyped using Illumina GSAv1.0, GSAv2.0, and GSAv2.0_EST arrays at the Core Genotyping Lab of the Institute of Genomics, University of Tartu. Samples were genotyped and PLINK format files were created using Illumina GenomeStudio v2.0.4. Individuals were excluded from the analysis if their call-rate was <95% or if their sex defined by heterozygosity of X chromosomes did not match their sex in the phenotype data. Before imputation, variants were filtered by call-rate <95%, HWE $p$-value < 1e−4 (autosomal variants only), and minor allele frequency <1%. Same analyses were conducted for association analysis and imputation of chromosome X, except for the HWE filter, which was not applied. Variant positions were updated to b37 and all variants were changed to be from the TOP strand using GSAMD-24v1-0_20011747_A1-b37.strand.RefAlt.zip files from the https://www.well.ox.ac.uk/~wrayner/strand/ webpage. Pre-phasing was conducted using Eagle v2.3 software[66] (number of conditioning haplotypes Eagle2 uses when phasing each sample was set to:—Kpbwt=20000) and imputation was done using Beagle v.28Sep18.793[67] with effective population size ne=20,000. The population specific imputation reference of 2297 whole genome sequencing (WGS) samples was used[68]. Association analysis was carried out using SAIGE (v0.38) software to implement a mixed logistic regression model with year of birth and 10 PCs as covariates in step I. In EstBB, SNVs with poor imputation quality (INFO score < 0.4) and minor allele count <5 were excluded from downstream association analysis.

**GWAS meta-analysis.** We conducted an inverse of variance weighted fixed-effects meta-analysis with single genomic control correction using GWAMA software (v2.2.2)[69]. A total of 40,542,692 variants were included in the meta-analysis of 28,086 women with POP and 546,291 female controls. We performed two tests of heterogeneity of effects across studies (Q-Cochran test and heterogeneity index ($I^2$)) calculation as implemented in GWAMA (v2.2.2)[69]. After meta-analysis we kept variants present in minimum 2 out of the 3 studies (totaling 15,065,244 variants) for down-stream analysis. Lead SNPs were identified as SNVs independent from each other with P-value less than or equal $5 \times 10^{-8}$. 500 kb was set to the maximum distance between LD blocks of independent significant SNPs to merge into a single genomic locus. Assuming a prevalence of 5% for symptomatic POP and an overall POP prevalence of 40% in the population, total-$h^2$ was estimated by single-trait LD score regression using the meta-analysis summary statistics and HapMap 3 LD-scores[19,70] and converted to the liability scale using LDSC v1.0.1 (https://github.com/bulik/ldsc). After excluding genome-wide significant loci (variants within $+/-500$ kb from lead signals), we calculated the SNP heritability of the remaining variants and the one corresponding to the significant loci.

**Gene prioritization criteria.** In order to move from genetic variants to plausible candidate genes, we used the following criteria. We prioritized candidate genes considering four main evidence levels: (1) nearest gene to the association peak; (2) genes containing shared causal variants and gene expression signatures unraveled by colocalization analyses; (3) genes containing coding variants or in high LD (r2 > 0.6) with these; (4) finally, we utilized the Mouse Genome Database[71] (http://www.informatics.jax.org) to evaluate the effect of candidate genes in mutant mice spanning embryo, growth/size/body, muscle, renal/urinary phenotypes.

**Colocalization analyses.** Colocalization analyses were conducted using COLOC (v.3.2.1) R package[62] and GWAS meta-analysis effect sizes and their variances. In the analysis, we compared our significant GWAS loci to all GTEXv8 and eQTL Catalog (https://www.ebi.ac.uk/eqtl/)[72] associations (excluding Lepik et al. 2017[73] and Kasela et al. 2017[74] due to sample overlap) within 1Mbp radius of a GWAS top signal. Prior probabilities were set to p1 = 1e−4, p2 = 1e−4 and p12 = 5e−6. Two signals were considered to colocalize if the posterior probability for a shared causal variant was 0.8 or higher.

**Gene-set analysis and tissue/cell-type expression analyses.** Gene-set analysis and tissue expression analysis were performed using MAGMA v1.08[12] implemented in FUMA v1.3.6a[14] and DEPICT[18], implemented in Complex-Traits Genetics Virtual Lab (CTG-VL 0.4-beta)[75].

In MAGMA v1.08, gene sets were obtained from Msigdb v7.0 for "Curated gene sets" and "GO terms". A total of 15,485 gene terms were queried. Tissue expression analysis was performed for 53 specific tissue types using MAGMA. DEPICT is an integrative tool that based on predicted gene functions highlights enriched pathways and identifies tissues/cell types where genes from associated loci are highly expressed.

**Genetic correlation.** The LDSC method and GWAS-MA summary statistics were used for testing genetic correlations[19] between POP, and data available for 561 traits in LD-Hub v1.9.3 (http://ldsc.broadinstitute.org) including traits from the following categories: lipids, smoking behavior, anthropometric, reproduction, cardiometabolic, and a range of traits from UKBB[20]. Cross-trait LD-score regression is not biased by sample overlap[76] and we accounted for multiple testing using a Bonferroni correction for 561 tests ($0.05/561 = 8.91 \times 10^{-5}$).

**Phenome-wide associations.** Pleiotropy was assessed comparing phenotype associations for the GWAS lead variants in two databases: PhenoScanner v2[22] using the *phenoscanner* (v1.0) R package (https://github.com/phenoscanner/phenoscanner), and GWAS Catalog (e96_r2019-09-24) implemented in FUMA[14]. GWAS catalog look-up also included variants in high LD with lead variants (r2 > 0.6). For visualization of results, a heatmap was created using the *pheatmap* library in R 3.6.1. and a modified script from (https://github.com/LappalainenLab/spiromics-covid19- eqtl/blob/master/eqtl/summary_phenoscanner_lookup.Rmd). The results obtained were filtered to keep one association per variant per trait, keeping studies from newer or larger studies. Descriptions of Experimental Factor Ontology (EFO) terms and classification of EFO broad categories were obtained from the GWAS Catalog. Missing categories were added by manually searching the EMBL-EBI EFO webpage (www.ebi.ac.uk/efo/).

**Derivation of PRS for POP.** In brief, PRS analysis requires two types of data: (1) base: summary statistics of genotype-phenotype associations at genetic variants genome-wide, and (2) target: genotypes and phenotype in individuals of an independent sample[77]. We constructed a POP PRS based on the summary statistics of the meta-analyses including IceUK and FinnGen, with 20,118 cases and 427,426 controls of European ancestry, leaving out EstBB as an independent target dataset.

Each PRS was computed for each woman in the EstBB ($N = 126,791$) by summing the product of the allele weighting and the allele dosage across the selected SNPs. We empirically evaluated a total of 19 different versions of PRS,

implementing two different methodologies: PRSice2 (v2.3.3)[24] and LDPred1.0.11[23], which use a clumping and thresholding and linkage-disequilibrium SNP-reweighting approach, respectively. Whilst PRSice2 automatically calculates and applies the PRS, in the case of LDPred1.0.11, STEROID (v0.1.1) tool was used for calculating PRS for all EstBB participants (https://genomics.ut.ee/en/tools).

**POP polygenic risk score calculation.** Genetic variants with MAF < 0.01, indels, and variants with imputation score 0.8 and lower in EstBB were removed from all polygenic risk score calculations. PRSice-2 uses a "clumping and thresholding" approach to clump genetic variants in close linkage disequilibrium[24], such that the remaining variants are independent of each other, and includes only those variants with a GWAS association P-value below a given threshold, with the threshold chosen to maximize the association of the risk score with POP. We tested the following thresholds: 1, 0.3, 0.1, 0.03, 0.01, 0.003, 0.001, 0.0003, and 0.0001, with a maximum LD between them set to r2 = 0.2. The number of SNPs included in each model are presented in Supplementary Data 11.

LDpred is a Bayesian approach that applies a continuous shrinkage model to modify effect sizes of SNPs to incorporate information on the strength of each variant's association in the GWAS and the underlying linkage disequilibrium structure[23]. To decrease the dimension of multicollinearity, SNPs were clumped with maximum LD between them set to r2 = 0.99. Then, 10 versions of PRSs were calculated by varying the fraction of causal SNPs on these values: Inf, 1, 0.3, 0.1, 0.03, 0.01, 0.003, 0.001, 0.0003, and 0.0001. Possible convergence issues were reported by the program for some fractions (different depending on the base study) while Gibbs sampler tried to estimate the posterior effect estimates.

**Criteria for discovery and validation set definition.** We divided the target dataset from EstBB into a discovery and validation dataset, according to their prevalent or incident status. The discovery dataset included 5379 prevalent cases and 21,516 controls. The selection of controls in the discovery set was randomized, including 4 controls per case. Since controls were defined as women who did not develop POP during follow-up (which initiates in first linkage to Estonian Health Insurance Fund in 26-11-2002 and ends in latest linkage to diagnoses dating from October 30-12-2019). Cases were not otherwise matched to controls. Therefore, controls tended to be younger than prevalent cases. In the discovery set, we tested all 19 PRS versions and selected the best PRS version for further analyses (Supplementary Fig. 1).

The validation set included 2517 incident cases and 96,109 controls, and in this set, we tested the predictive ability of PRS (Supplementary Fig. 1). The validation set was further filtered to a validation subset, where only incident cases and controls, which presented little or no missing data of clinical risk factors data in EstBB were kept. This included a total of 2104 incident cases and 24,753 controls, where scores were tested alone or in combination with clinical variables (Supplementary Fig. 1). While it is true that the number of cases slightly differs between validation set and subset, approximately four-fold more controls increase the power of our analysis and results in more precise effect estimates when evaluating the predictive ability of the PRS alone in the validation set.

**Selection of best PRS model.** The discovery set was used in the initial analyses in order to select the best predicting PRS version through a logistic regression model adjusted for age, age squared, first 10 principal components, and batch effects. The model that offered the smallest p-value towards case-control discrimination was selected for further analyses.

**Predictive ability of the PRS.** We standardized the best PRS version and also categorized it into different percentiles (<5%, 5–15%, 15–25%, 25–50%, 50–75%, 75–85%, 85–95%, >95%). Survival modeling and Cox proportional hazard models were used to estimate the Hazard Ratios (HR) corresponding to one standard deviation (SD) of the continuous PRS for the validation set. Harrell's C-statistic was used to characterize the discriminative ability of each PRS. Cumulative incidence estimates were computed using Kaplan–Meier method. We used survival modeling, where age was used as a timescale to properly account for left-truncation in the data and right-censoring. Cox proportional hazard models were also used as specified before to assess the differences between genetic risk in different age strata (women <40 years old (y), 40–50 y, 50–60, 60–70 y, >70 y).

**Predictive ability of PRS and classical risk factors.** Next, we use the validation subset to assess the predictive ability of PRS and five clinical risk factors (number of children, BMI, ever smoked, asthma, and constipation) alone or in combination, and clinical risk factors together with PRS. Clinical risk factors were chosen according to literature mining and data availability in Estonian Biobank. Information on the number of children, BMI, and smoking were extracted from questionnaire data, whereas ICD10 codes J45 and K59.0 were used for asthma and constipation, respectively. Number of births was chosen as a clinical covariate because it captures more than one delivery event as well as providing the possibility to capture the weakening of the pelvic floor already happening during pregnancy and excluding possible first trimester miscarriages (which are not affected by extra weight and weakness of pelvic floor structure and thus are unlikely to contribute to prolapse development).

**Reporting summary**. Further information on research design is available in the Nature Research Reporting Summary linked to this article.

## Data availability

The full meta-analysis summary statistics generated in this study have been deposited in the GWASCatalog (https://www.ebi.ac.uk/gwas/) under accession code GCST90102470. The PRS summary statistics generated in this study have been deposited in the PGS Catalog (https://www.pgscatalog.org/) database under accession code PGS002288. The individual level data from Estonian Biobank are available under restricted access for containing sensitive information from healthcare registers, access can be obtained through the Estonian biobank upon submission of a research plan and signing a data transfer agreement. All data access to the Estonian Biobank must follow the informed consent regulations of the Estonian Committee on Bioethics and Human Research, which are clearly described in the Data Access section at https://genomics.ut.ee/en/content/estonian-biobank. A preliminary request for raw genetic and phenotype data must first be submitted via the email address releases@ut.ee. Icelandic and UKBB summary statistics can be accessed from http://www.decode.com/summarydata and FinnGen summary statistics can be downloaded after filling this form (https://elomake.helsinki.fi/lomakkeet/102575/lomake.html). We queried mouse mutant phenotypes utilizing Mouse Genome Database (MGI6.18, latest update from 05/04/2022) (http://www.informatics.jax.org/).

## Code availability

The following software packages were used for data analysis: In EstBB, GenomeStudio (v2.0.4), Eagle (v2.3), and Beagle (v28Sep18.793) were used as part of the standard genotyping and imputation pipeline. EstBB GWAS was carried out with SAIGE (v0.38). GWAS meta-analysis was conducted using the GWAMA software (v2.2.2). For colocalization, COLOC (v.3.2.1) was used. FUMA v1.3.6a was used for GWAS catalog (e91_r2018-02-06) look-up. For gene set and tissue/cell type specific enrichment analysis MAGMA v1.08 implemented in FUMA v1.3.6a and DEPICT v1 implemented in CTG-VL O.4-beta were used. Msigdb v7.0: Gene sets were obtained from Msigdb v7.0 for "Curated gene sets" and "GO terms" as implemented in FUMAv1.3.6. ggplot2 package was used for visualization of results. ANNOVARv2019Oct24 was used as implemented in FUMA v1.3.6a. LDHub v1.9.3 was used for genetic correlation analysis and LDSCv1.0.1 was used for SNP-heritability calculations (https://github.com/bulik/ldsc). PhenoScanner v2 was used for look-up of phenotype associations for the GWAS lead variants in previous GWAS studies, using the *phenoscanner* (v1.0) R package, and the results were visualized using *pheatmap* library in R 3.6.1. and a modified script from (https://github.com/LappalainenLab/spiromics-covid19-eqtl/blob/master/eqtl/summary_phenoscanner_lookup.Rmd). To calculate the polygenic risk scores, LDpred1.0.11, STEROID v0.1.1 (https://genomics.ut.ee/en/tools) and PRSice 2 v2.3.3 were used. Survival analysis were conducted using *survival* package in R 3.6.1. All other analyses were conducted in R 3.6.1. We queried mouse mutant phenotypes utilizing Mouse Genome Database (MGI6.18, latest update from 05/04/2022) (http://www.informatics.jax.org/).

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

## Acknowledgements

NPG was supported by MATER Marie Sklodowska-Curie which received funding from the European Union's Horizon 2020 research and innovation program under grant agreement No. 813707. K.L., T.L., M.L., and R.M. are supported by the Estonian Research Council grant PRG687. This study was supported by the European Union from the Horizon 2020 grant INTERVENE. This study was funded by European Union through the European Regional Development Fund Project No. 2014-2020.4.01.15-0012 GEN-TRANSMED. Computations were performed in the High Performance Computing Center, University of Tartu. T.L.P., R.A., and H.R. are supported by the Academy of Finland grants no 315921 and 321763 and Sigrid Juselius foundation. We want to acknowledge the participants and investigators of the Icelandic, FinnGen, UKBB, and EstBB studies. The Genotype-Tissue Expression (GTEx) Project was supported by the Common Fund of the Office of the Director of the National Institutes of Health, and by NCI, NHGRI, NHLBI, NIDA, NIMH, and NINDS. The data used for the analyses described in this manuscript were obtained from the GTEx Portal on 10/05/21. The funders had no role in study design, data collection and analysis, decision to publish, or preparation of the manuscript.

## Author contributions

T.L. and R.M. conceived the idea of the study. N.P.G. performed the meta-analysis, GWAS follow-up, and PRS analyses. T.L., R.M., and K.L. supervised analyses. M.L. performed the colocalization analyses. T.T.P, H.R.R., and R.K.A. provided clinical context and clinical interpretation of findings. Estonian Biobank Research Team collected and provided EstBB data. N.P.G. wrote the first draft of the manuscript. All authors critically reviewed the paper.

## Competing interests

The authors declare no competing interests

## Additional information

## Estonian Biobank Research Team

Andres Metspalu[1,3], Mari Nelis[1], Lili Milani[1], Tõnu Esko[1] & Georgi Hudjashov[1]

[3]Institute of Cell and Molecular Biology, University of Tartu, Tartu, Estonia.

