## [Peer Review File · Nature Communications]

Advancing our understanding of genetic risk factors and potential personalized strategies for pelvic organ prolapseREVIEWER COMMENTS

Reviewer #1 (Remarks to the Author):

This is a nice GWAS meta-analysis of a trait that, like many female reproductive traits, is under-researched. The authors combined data from a previously published GWAS for pelvic organ prolapse (PO), with data from the Estonia Biobank and discovered 19 new genetic signals in addition to replicating the previous 7. They went on to test how a polygenic risk score (PRS) could predict risk of POP and how addition of clinical risk factors improved the predictive power. In general, the methodology is sound and the findings robust. I think there are a couple of areas that it would be useful to provide more clarity particularly for the non-specialist reader and I have the following specific suggestions:

1. It is not always clear in the text whether the results are referring to the new genes identified or to the previously reported GWAS hits. It might be helpful to make it clear in figure 2 which are the new ones. Clearer discussion about exactly what this study is adding to our knowledge over and above the previous study would be helpful. For example, line 253 talks about 'candidate genes' are these genes that have previously been reported in experimental models or come from GWAS or both?
2. Regarding figure 2, why are the loci not in order (minor point, but seemed odd)? and there are 24 loci in that figure, but you report 26 in the results?
3. It would be good to have a more complete results table in the supplementary data, showing the associations in the three datasets, plus linking directly to the genes highlighted in Figure 2.
4. Some of the Figure legends could be clearer, eg Fig3, it isn't stated which study this is or any numbers of individuals and because you don't read the methods until the end of the paper, it would be useful to have some more of these details in the main text.
5. There were no signals on the X chromosome. Was data for the X available for all studies?
6. For the PRS work I think the use of the validation set and then the subset adds a level of confusion and you only present results for the subset, so not sure why you need the slightly larger set (and it is only slightly larger with respect to cases)
7. How were controls selected for the PRS study? It would be very helpful to have some summary stats for the cases and controls used in the PRS. You say that age is younger in controls, why were controls not matched for age? Also, it might be interesting to exclude controls that didn't have any ICD10 codes, as these might not have had the opportunity for case status to be recorded. Were ICD10 codes the primary code or could they be secondary to the main diagnosis?
8. Oestrogen exposure seems to be an important mechanism here. Could you investigate the role of this in your data either directly or even indirectly, eg. through effect of reproductive lifespan or HRT use?
9. I don't think it is appropriate to claim this is the first PRS for a female specific trait, PRS for menarche and menopause have been previously reported for example.
10. Lines 313-318, I think this paragraph needs to be clarified, I did not follow the arguments made.
11. For the supplementary tables it would be helpful to have meaningful labels to each column rather than the shorthand notation

Reviewer #2 (Remarks to the Author):

The authors carried out a genome-wide meta-analysis of pelvic organ prolapse (POP) using data from several population based studies, doubling the previous sample size and increasing the number of genetic loci from 7 to 26. This is an interesting and well-written paper that adds to the understanding of the genetic basis of this condition, though these new signals will require future replication. Up-to-date methodology is then employed to identify relevant biology, correlations with other phenotypes and predictive models. Predictive models have been tested robustly, which I think is a particular strength. I have the following minor comments:

1. Table 1. It would be helpful to add gene annotations and also give the chr and pos of the signals to make it easy to cross reference with other tables and the text, for example, with Supp Fig 2.
2. Gene prioritization: If the genes considered had to show regulatory effects, did this mean that genes without such data (e.g. due to missing data) were not considered?
3. Genetic correlations: For the traits identified with significant genetic correlations, which studies were these from and did they overlap with your meta-analysis sample?
4. Genetic correlations: Was the timing of hysterectomy in relation to POP considered – I think this can be a cause of but also in some cases a treatment for prolapse? It could be interesting to explore this further.
5. Prediction: Was age included as a variable in all prediction models and what effect does this have on the C-statistics over PRS?
6. Prediction: How were the clinical risk factors chosen for the prediction model?
7. Discussion: I think the following statement should be more cautious: “this is the first study evaluating a PRS for a female-specific non-cancer trait” - there are other studies employing similar methods e.g. Rafnar et al. Nat Comms 2018 doi:10.1038/s41467-018-05428-6; Ruth et al. Nature 596 2021 doi: 10.1038/s41586-021-03779-7; Joo YY et al. J Clin Endocrinol Metab. 2020 doi: 10.1210/clinem/dgz326.
8. Methods: Some further detail is required to describe how the independent significant loci were identified and any MAF and imputation threshold applied to the GWAS results.
9. Methods: Please clarify the meaning of “poor effect first trimester miscarriages likely have in prolapse”.
10. Supplementary Figure 3 and 4 should be Supplementary Figure 4 and 5, etc.

REVIEWER COMMENTS

Reviewer #1 (Remarks to the Author):

This is a nice GWAS meta-analysis of a trait that, like many female reproductive traits, is under-researched. The authors combined data from a previously published GWAS for pelvic organ prolapse (PO), with data from the Estonia Biobank and discovered 19 new genetic signals in addition to replicating the previous 7. They went on to test how a polygenic risk score (PRS) could predict risk of POP and how addition of clinical risk factors improved the predictive power. In general, the methodology is sound and the findings robust. I think there are a couple of areas that it would be useful to provide more clarity particularly for the non-specialist reader and I have the following specific suggestions:

1. It is not always clear in the text whether the results are referring to the new genes identified or to the previously reported GWAS hits. It might be helpful to make it clear in figure 2 which are the new ones. Clearer discussion about exactly what this study is adding to our knowledge over and above the previous study would be helpful. For example, line 253 talks about ‘candidate genes’ are these genes that have previously been reported in experimental models or come from GWAS or both?

We thank the reviewer for their suggestions to clarify some aspects of the manuscript, which are further addressed below.

Firstly, we have reframed the results and discussion to clarify which findings correspond to previously reported GWAS hits or the novel ones from the current work, which we hope at the same time adds clarity of what this study adds in comparison to the previous one.

Results

Line 136 “Overall, our results are in line with previous GWAS findings, replicating and highlighting associations near WNT4, EFEMP1, FAT4, IMPDH1, TBX5, SALL1.”

Line 143 ‘Data from mouse models also supported the roles of potential novel candidate genes such as ACADVL (Acadvltm1Vje/Acadvltm1Vje), PLA2G6 (Pla2g6m1Sein/Pla2g6m1Sein) and HOXD13 (Hoxd13tm1Ddu/Hoxd13+)’

Line 148 “Additionally, we confirmed previously reported gene candidates with substantial evidence from mice model studies such as LOXL1 and EFEMP1.”

Line 154 “Based on functional impact, we were able to identify three novel candidate genes for POP.”

Discussion

Line 268-270 “Additionally, we further support previous findings highlighting associations near WNT4, EFEMP1, FAT4, IMPDH1, TBX5, SALL1. Beyond EFEMP1, we further propose several novel candidate genes”

Line 276 “on the novel locus described in chromosome 21, the most plausible candidate gene was ADAMTS5. ”

Line 281 “Our study also reinforces urogenital development as a key process in the pathogenesis of POP highlighting novel genes such as DVL2, WT1, HOXD13. ”

Secondly, we have now updated the plot (Figure 2) accordingly, highlighting the novel genes in yellow frames and the previously reported in grey frames. Please see the new Figure at the end of this comment's answer.

Thirdly, we have also reframed the discussion to provide a clearer review of which knowledge our study adds above the previous study, which have been added as disclosed below. We hope that a clearer distinction between reported and novel genes through the text as addressed in the first part of this comment also helps the reader to identify the novelty of this study in the text. In summary, the main added value of our study is the identification of 19 novel loci, extensive follow-up of GWAS signals and data-driven gene prioritisation approach. Additionally, in this study we developed and evaluated the first polygenic risk score for POP.

Results, lines 138-141

Colocalization analyses allowed us to highlight an additional candidate gene, LDAH in the previously reported 2p16.1 locus. Additionally, significant colocalization signals allowed us to prioritise potential novel candidate genes such as VCL, CHRDL2, DUSP16, LOXL1-AS1, CRISPLD2, KLF13, ADAMTS5 and MAFF.

Discussion lines 255-256: "In contrast to previous efforts characterizing genome-wide signals underlying POP¹², we used a combination of different data layers to map potential candidate genes, which provide novel insights into the biology of prolapse development, open up new avenues for further functional studies, and identify links with other health outcomes, which might have important implications for patient management and counselling."

Discussion, lines 310-320:

"Here we present a comprehensive GWAS follow-up which culminated in defining possible convergence in meaningful biological pathways through defined test sets of genes such as "Connective Tissue Development" ($p=2.01 \times 10^{-6}$), "Chondrocyte differentiation" ($p=6.63 \times 10^{-4}$), and "In utero embryonic development" ($p=4.91 \times 10^{-8}$) and link these to specific tissues and cell types such as "Cervix/ectocervix" ($p=1.3 \times 10^{-5}$), "Uterus" ($p=1.5 \times 10^{-5}$), "Embryoid bodies" ($p=8.6 \times 10^{-6}$) and "Smooth muscle" ($p=7.3 \times 10^{-4}$). These findings might aid in choosing the relevant tissue or cell type for in-vitro experiments to further elucidate molecular mechanisms underlying the genome-wide significant loci identified."

The added value of our study in comparison to previous efforts¹² is best highlighted by presenting the first polygenic risk score in POP, which substantially advances the concept of using genomic information to stratify women in gynecological conditions."

Discussion line 274: "Additionally, our study highlights a plausible role of A Disintegrin and Metalloprotease with Thrombospondin Motifs (ADAMTS) in POP"

Discussion line 301 "many associations we report point towards a link between metabolic and cardiovascular health and pelvic organ prolapse (KLF13, DUSP16, MAFF, VCL and LDAH), which is a novel and particularly interesting genetic association unraveled by our study"

Fourthly, we have reformulated and clarified in line 253 (now line 270) that the candidate genes refer to those novel candidate genes from our study, coming from GWAS after

conducting gene prioritization. “We further propose several **novel** candidate genes (CHRDL2, ACADVL, PLA2G6)”

2. Regarding figure 2, why are the loci not in order (minor point, but seemed odd)? and there are 24 loci in that figure, but you report 26 in the results?

In the revised manuscript we have sorted the loci in Figure 2, presenting in the first panel the ones that have been previously reported and in second and third panels the novel ones highlighted in yellow. We have added the two additional loci which were not present in the previous figure, corresponding to the two loci in chromosome 4 (4q28.1 and 4q13.2) (see New Figure 2 in response to comment 1).

3. It would be good to have a more complete results table in the supplementary data, showing the associations in the three datasets, plus linking directly to the genes highlighted in Figure 2.

We have now added a more complete results table in the supplementary data. In the revised version of the manuscript this can be found as Supplementary Data 1, showing the associations in the three datasets (firstly showing reported loci and secondly novel findings as in Table 1 and Figure 2), plus linking directly to the genes highlighted in Figure 2.

4. Some of the Figure legends could be clearer, eg Fig3, it isn't stated which study this is or any numbers of individuals and because you don't read the methods until the end of the paper, it would be useful to have some more of these details in the main text.

We have now rephrased the following figure legends, which we hope provides some more methodological background earlier in the text:

Results, line 190; “Figure 3. Genetic correlation analyses. The genome-wide genetic correlation of POP GWAS meta-analysis summary statistics with other phenotypes was estimated using published GWASs (43 phenotypes) and GWASs of UK Biobank data (518 phenotypes), available in LD-Hub (see study source in Supplementary Data 7). Significant

genetic correlations reveal overlap of genetic risk factors for POP across several groups of traits (grouped by colours): anthropometric, cardiometabolic, ICD10 diagnoses, job type, reproductive traits and self-reported conditions. Center values show the estimated genetic correlation (r_g), which is presented as a dot and error bars indicate 95% confidence limits.”

Results, line 235; “**Figure 4: Cumulative incidence by PRS categories.** Cumulative incidence of POP in % scaled by age in the validation set of Estonian Biobank (2,517 incident cases and 96,109 controls) for different POP PRS percentiles (<5%, 5%-15%, 15%-25%, 25%-50%, 50%-75%, 75%-85%, 85%-95%, >95%). Survival modeling and Cox proportional hazard models were implemented, using age as a time scale for properly accounting for left-truncation and right-censoring in the data.”

Results, line 241: “**Figure 5. Predictive ability of PRS and clinical variables in incident status.** Green dots represent polygenic risk score (PRS), orange dots represent five established risk factors and purple dots represent genetic and/or clinical combined models C-stat indexes. Cox proportional hazard models were used and age was used as a time scale for properly accounting for left-truncation and right-censoring in the data. A) C-stat for clinical variables and PRS alone or in combination in the validation subset of Estonian Biobank (2,104 cases and 24,753 controls). B) C-stat adjusted by batch effects and 10 first principal components in the validation subset of Estonian Biobank (2,104 cases and 24,753 controls).”

We also added more context in some Results sections, which we hope contextualizes better the work done, since Methods are presented later in the text.

Results, line 91. Genome-wide inference. We performed a meta-analysis with data from three studies (Icelandic and UKB cohorts, FinnGen study and EstBB)

Results, line 169. “**Genetic correlation.** Genetic correlation with POP was estimated through pairwise comparison with published GWASs (43 phenotypes) and GWASs of UK Biobank data (518 phenotypes), totaling 561 phenotypes,”

line 210 This version showed an $OR=1.42$ (1.37 to 1.47) and $p=2.59 \times 10^{-89}$ towards the case-control discrimination in the discovery set (including 5,379 prevalent cases and 21,516 controls)

line 212 We proceeded to test the predictive ability of the PRS in the validation set (totaling 2,517 incident cases and 96,109 controls).

line 224-227: From the validation set of EstBB we further selected a validation subset of 2,104 cases and 24,753 controls who had little or almost no missing clinical covariate data (Supplementary Figure 1, Supplementary Data 12), which allowed us to test the predictive ability of the PRS alone or in combination with 5 clinical variables (number of children, body mass index (BMI), ever smoked, constipation and asthma).

5. There were no signals on the X chromosome. Was data for the X available for all studies?

We thank the reviewer for pointing this out. Unfortunately, the imputation for chromosome X from the Estonian Biobank cohort was not available at the time of the analysis. We have now rerun the association analysis and meta-analysis including X chromosome for all cohorts, which shows no significant associations with pelvic organ prolapse. We have clarified this and modified text accordingly in Supplementary Methods, line 34-35:

“Same analysis were conducted for association analysis and imputation of chromosome X, except for the HWE filter, which was not applied.”

We also have rerun and updated the summary statistics association results including chromosome X for all studies in the meta-analysis (see Data availability section).

6. For the PRS work I think the use of the validation set and then the subset adds a level of confusion and you only present results for the subset, so not sure why you need the slightly larger set (and it is only slightly larger with respect to cases)

Whilst we agree that the number of cases slightly differs (from 2,517 incident cases in the validation set to 2,104 in the validation subset) there is a large difference in the number of controls (96,130 vs 24,780). The only reason why a validation subset was defined is due to larger availability of clinical risk factors in this subset (as specified in Supplementary Data 12). However, we believe the evaluation of the predictive ability of the PRS in a larger set (validation set) is justified since adding approximately 4-fold more controls results in increased power and more precise effect estimates of risk stratification using the polygenic risk score. For instance, we showed a hazard ratio (HR) of 1.63 (95% CI 1.37-1.93) when comparing women in the top5% of genetic risk versus all the rest of women in the validation set. The same comparison in the validation subset results in a HR of 1.69 (95% CI 1.14-2.02). We are prone to maintain the two validation groups (full set and subset) since it offers the best estimate possible for genetic risk alone and might be of interest for future comparisons in replication cohorts/studies which solely assess genetic risk predictive ability in POP development.

We believe the two sets are worth keeping and hope to provide a clearer justification and distinction between validation set and subset in the manuscript.

To this end, we have divided the section “Predictive ability of the PRS and classical risk factors” into two different sections:

- 1) Line 481, ‘Predictive ability of the PRS’ (which was done using the validation set)*
- 2) Line 491 ‘Predictive ability of the PRS and classical risk factors’ (which was done using the validation subset)*

And added the following justification in Methods, line 472-475:

“While it is true that the number of cases slightly differs between validation set and subset, approximately 4-fold more controls increase the power of our analysis and results in more precise effect estimates when evaluating the predictive ability of the PRS alone in the validation set.”

7. How were controls selected for the PRS study? It would be very helpful to have some summary stats for the cases and controls used in the PRS. You say that age is younger in controls, why were controls not matched for age? Also, it might be

interesting to exclude controls that didn't have any ICD10 codes, as these might not have had the opportunity for case status to be recorded. Were ICD10 codes the primary code or could they be secondary to the main diagnosis?

We hope the explanations below clarify the questions raised. In the PRS analysis, two main analyses were conducted, 1) selection of the best PRS amongst 19 PRS versions in the discovery set, where a logistic regression adjusted by age, age squared, batch effects and first 10 principal components. and 2) testing the predictive ability of the PRS in the validation set, where Cox proportional hazard models were built, using age as a time scale. Below we break down and address the different points raised in the comment:

1.1. How were the controls selected for the PRS study? It would be very helpful to have some summary stats for the cases and controls used in the PRS.

The selection of controls in the discovery set was randomized, including 4 controls per case (5,379 prevalent cases and 21,516 controls). In this set, controls were defined as women who did not develop pelvic organ prolapse during follow-up (referring to the last time point when genetic data from EstBB is linked to diagnoses information from the national health-related registries). For that reason controls tended to be younger than prevalent cases. In the present manuscript we have added a supplementary table (Supplementary Data 11) showing the mean and standard error of age and total N of women by different age strata between all case-control groups defined in Estonian Biobank analyses.

1.2. You say that age is younger in controls, why were controls not matched for age?

Whilst it is true that age differs between cases and controls, the logistic regression was adjusted for age and age squared in the discovery set and it still has meaning because its main aim is to select the best fit version of the PRS, amongst the 19 we constructed, and keep with this for further analyses which test predictive ability of the PRS. In the validation set analyses of the PRS, the effect of age was controlled by using age as the time scale in Cox proportional hazard models, thus comparing only women from same ages and avoiding inflation in prediction C-statistics solely due to age difference between groups.

The fact that in the discovery set we included 4 controls per each case, totaling a number of 21,516 controls, supports the use of unconditional (rather than conditional) logistic regression because the analysis strata is not very small for each age stratum (see Supplementary Data 11) therefore problems of sparse data are unlikely to occur (PMID: 3766505/, PMID: 26916049/)

1.3. Controls who have not had any ICD10 code registered: *We thank the reviewer for pointing out this observation. In total, there are 30 controls which have not had any health system information source in our biobank. We have removed them from all analyses (all 30 controls are part of the validation set), and modified sample sizes accordingly throughout the text and Supplementary Data 11 and 12.*

1.4. ICD-10 codes as primary or secondary to the main diagnosis: *ICD-10 codes in Estonia can be either primary or secondary to the main diagnosis, but since we are interested in capturing all women with N81 diagnosis, this distinction has little effect in our analyses. Reassuringly, the prevalence in each cohort closely reflects the prevalence estimates for symptomatic POP across the general population (3–6%) (PMID: 18799443) being 2,5% in Icelandic dataset, 5,2% in UKBB, 12,7% in FinnGen and 6,2% in EstBB. Whilst in FinnGen, Icelandic data and UKBB the diagnoses are obtained from in-hospital registers, in EstBB information on diagnoses was obtained by general practitioners (GPs), who were involved as recruiters for a subset of the cohort (first recruitment of 50K individuals), and in further sets diagnoses were extracted through yearly linkages with the Estonian Health Insurance Fund*

(EHIF) which offers an extraordinary coverage and representation of the Estonian health care situation. Accordingly, the EHIF is the only organization in Estonia dealing with compulsory health Insurance, covering 95.5% of the population (99% in EstBB cohort), which firmly supports an accurate representation of the whole Estonian health care situation (https://www.haigekassa.ee/sites/default/files/Maailmapangauuring/veeb_eng_summary_report_hk_2015_mai.pdf).

Additionally, in the Estonian Biobank, we estimated that nearly half (48.9%) of the women defined as women with pelvic organ prolapse in our study already have multiple diagnosis counts (meaning these are not random) and this number will likely increase with future linkages.

All in all, we hope the explanations above clarify these aspects. We have added a more complete summary statistics information in regard to the cases and controls utilized in the polygenic risk score analyses in Supplementary Data 11 and modified N of controls through the text and in Supplementary Figure 1.

8. Oestrogen exposure seems to be an important mechanism here. Could you investigate the role of this in your data either directly or even indirectly, eg. through effect of reproductive lifespan or HRT use?

We thank the reviewer for raising this interesting question. Due to larger sample size availability, we believe that genetic correlation analysis offer a better approach than using individual level data from Estonian Biobank, which retrieves through questionnaire self-reported current/former usage of HRT and self-reported age at menarche and menopause, and thus only answered by a subset of participants.

Now we have presented the shared genetic background estimates of pelvic organ prolapse and three traits which offer an indirect measure of oestrogen exposure, as follows:

- 1) Ever had hormone-replacement therapy (HRT), which showed a moderate and nominally significant genetic correlation ($rg=0.15, se=0.05, p=0.002$) with pelvic organ prolapse (we used summary statistics from UKBB available in <http://www.nealelab.is/uk-biobank/>),
- 2) Age at menarche, which showed a small negative correlation with POP ($rg=-0.08, se=0.03, pval=0.006$) (source of the study: PMID:25231870).
- 3) Age at menopause, which showed no correlation with pelvic organ prolapse ($rg=-0.05, se=0.04, p=0.28$). (source of the study: PMID:26414677)

These results are now included in Supplementary Data 7.

Our data suggests a positive correlation between the usage of HRT and POP, although not reaching corrected significance. Since HRT tend to be prescribed after oophorectomy/hysterectomy (PMID: 27793381, 27716751/) which might be also leading to pelvic organ prolapse, this might provide an alternative explanation of the association trend and it is difficult to entangle correlation from causation. In this line, studies on the effect of HRT in POP show conflicting results (PMID:22453694/, 12738142, 15746674, 15339758, 19384125, 21603077).

We hypothesize that the small negative correlation between age at menarche and pelvic organ prolapse is likely explained by the shared genetic background between age at first birth and age at menarche which has been consistently described before (PMID: 34211149/)

(PMID:27798627). In our study, age of first birth shows a moderate correlation with pelvic organ prolapse ($r_g = -0.19$, $p = 1.42 \times 10^{-7}$), which might explain this association with age at menarche rather than capturing the effects of oestrogen exposure.

We added all genetic correlation results for the 561 traits tested in Supplementary Data 7, and take the reviewer's suggestion as an interesting lead for future research:

Line 289-291: "Future studies assessing sex-specific oestradiol exposure with larger sample sizes and more scaled time points, might inform the direction of association between oestrogen lifetime exposure and the development of POP. "

9. I don't think it is appropriate to claim this is the first PRS for a female specific trait, PRS for menarche and menopause have been previously reported for example.

We agree with the reviewer and have revised this sentence as suggested in the following lines:

Line 318: "The added value of our study in comparison to previous efforts¹² is best highlighted by presenting the first polygenic risk score in POP, which substantially advances the concept of using genomic information to stratify women in gynecological conditions."

Line 369. "In summary, this study is the first to explore polygenic risk scores as a tool to inform risk stratification strategies in POP and highlights the potential to improve predictive ability when adding genetic risk on top of clinical risk factors".

10. Lines 313-318, I think this paragraph needs to be clarified, I did not follow the arguments made.

We have now added new sentences in this paragraph which we hope better convey the point we wanted to express, now these lines correspond to lines 346-348.

"We observed no heterogeneity of effects between cohorts, which on one hand adds reliability to the identified genetic associations and on the other hand supports leveraging electronic health records data to ascertain this phenotype."

11. For the supplementary tables it would be helpful to have meaningful labels to each column rather than the shorthand notation

We have now added meaningful labels in the columns of every supplementary table. Please note in the present manuscript these are referred as Supplementary Data 1 to 13.

Reviewer #2 (Remarks to the Author):

The authors carried out a genome-wide meta-analysis of pelvic organ prolapse (POP) using data from several population-based studies, doubling the previous sample size and increasing the number of genetic loci from 7 to 26. This is an interesting and well-written paper that adds to the understanding of the genetic basis of this condition, though these new signals will require future replication. Up-to-date methodology is then employed to identify relevant biology, correlations with other phenotypes and predictive models. Predictive models have been tested robustly, which I think is a particular strength. I have the following minor comments:

1. Table 1. It would be helpful to add gene annotations and also give the chr and pos of the signals to make it easy to cross reference with other tables and the text, for example, with Supp Fig 2.

We have now added chromosome, position and added the gene(s) showcased in Figure 2, in Table 1.

Table 1. Results for the genome-wide significant index variants in the 26 loci associated with POP identified in the GWAS meta-analysis of 28,086 women with POP and 546,291 female controls.

Reported associations

Chromosome	Position ^a	Locus	Index variant (Effect Allele)	Effect Allele Frequency	p-value	Odds Ratio (95% CI)	Q-Cochran p-value	Prioritised gene(s)
1	2245682 15	1p36.12	rs3820282(T)	0.16	3.94×10^{-31}	0.86 (0.84-0.88)	0.014	WNT4
2	5610274 4	2p24.1	rs11899888(G)	0.11	4.01×10^{-16}	1.11 (1.09-1.14)	0.819	EFEMP1
2	2087810 5	2p16.1	rs9306894(G)	0.38	5.61×10^{-24}	1.10 (1.08-1.12)	0.298	GDF7, C2orf43
2	5611130 9	2p24.1	rs3791675(T)	0.25	1.23×10^{-13}	0.92 (0.90-0.94)	0.190	EFEMP1
4	1269246 84	4q28.1	rs28403275(C)	0.18	1.58×10^{-22}	1.12 (1.10-1.15)	0.164	FAT4

7	1280328 33	7q32.1	rs72624976(T)	0.01	1.14×10^{-9}	0.84 (0.79- 0.89)	0.002	IMPDH1
12	1146734 21	12q24.21	rs1247943(A)	0.54	1.68×10^{-21}	0.91 (0.90- 0.93)	0.195	TBX5
16	5148812 9	16q21.1	rs12325192(T)	0.18	1.14×10^{-21}	0.89 (0.87- 0.91)	0.046	SALL1

Novel associations

Chr	Position ^a	Locus	Index variant (EA)	EAF	p-value	OR (95% CI)	Q-Cochran p-value	Prioritised gene(s)
2	1769216 44	2q31.1	rs77648136 (T)	0.15	4.81×10^{-8}	0.93 (0.91- 0.96)	0.607	HOXD13
3	1277213 33	3q21.3	rs58170120 (A)	0.18	1.17×10^{-10}	1.08 (1.06- 1.11)	0.186	SEC61A1
4	6666689 5	4q13.2	rs201194999 (T)	0.30	2.42×10^{-8}	0.89 (0.85- 0.93)	0.508	EPHA5
4	1270761 88	4q28.1	rs10013769 (G)	0.65	1.26×10^{-10}	1.07 (1.05- 1.09)	0.876	FAT4

5	4978695	5p15.32	rs42400 (G)	0.36	4.22×10^{-11}	0.94 (0.92-0.96)	0.810	ADAMTS16
5	1275120 64	5q23.3	rs251217 (G)	0.61	1.31×10^{-10}	1.06 (1.05-1.08)	0.982	SLC12A2, FBN2
8	7190558 7	8q13.2	rs1493202(G)	0.52	3.56×10^{-8}	1.05 (1.03-1.07)	0.779	LACTB2- AS1
9	1802397 7	9p22.3	rs10810888(G)	0.65	4.00×10^{-8}	1.05 (1.03-1.07)	0.874	ADAMTSL1
10	7636882 3	10q22.1	rs10762631(A)	0.09	3.76×10^{-8}	0.92 (0.89-0.95)	0.276	VCL
11	7439251 4	11q13.4	rs4944936 (C)	0.72	7.13×10^{-12}	0.93 (0.91-0.95)	0.589	CHRD2
11	1030899 1	11p15.4	rs6484161 (T)	0.31	5.89×10^{-9}	1.06 (1.04-1.08)	0.190	SBF2,ADM
11	3247980 7	11p13	rs11031796 (A)	0.31	2.47×10^{-15}	0.93 (0.91-0.94)	0.002	WT1
11	3234639 7	11p13	rs35166569 (C)	0.09	2.54×10^{-8}	0.93 (0.91-0.95)	0.078	WT1

12	1266841 0	12p13.2	rs12314243 (T)	0.12	3.66×10^{-9}	1.09 (1.06- 1.12)	0.479	DUSP16
12	1146699 96	12q24.21	rs73197353 (C)	0.08	1.63×10^{-8}	1.12 (1.08- 1.17)	0.340	TBX5
15	3302348 6	15q13.1	rs12915554 (A)	0.32	1.06×10^{-8}	0.95 (0.93- 0.96)	0.875	GREM1
15	3163642 4	15q13.2	rs4779517 (G)	0.49	1.10×10^{-11}	1.07 (1.05- 1.09)	0.088	KLF13
15	7422538 8	15q24.1	rs4886778 (A)	0.46	4.12×10^{-8}	1.05 (1.03- 1.07)	0.029	LOXL1
16	8494047 9	16q24.1	rs1874008 (C)	0.77	5.77×10^{-9}	0.94 (0.92- 0.96)	0.046	CRISPLD2
17	7129898	17p13.1	rs72839768 (A)	0.02	4.66×10^{-9}	1.19 (1.12- 1.26)	0.096	DVL2, ACADVL
21	2863668 4	21q21.3	rs235929 (C)	0.39	2.01×10^{-12}	0.93 (0.92- 0.95)	0.884	ADAMTS5
22	3859823 4	22q13.1	rs2267372 (G)	0.61	1.07×10^{-13}	0.93 (0.91- 0.95)	0.299	MAFF,PLA 2G6

^aPositions are according to GRCh37.

2. Gene prioritization: If the genes considered had to show regulatory effects, did this mean that genes without such data (e.g. due to missing data) were not considered?

We appreciate and agree with the reviewer's viewpoint that only considering regulatory data might result in dismissing other plausible candidate genes for pelvic organ prolapse that do not have regulatory data available. Certainly, studies assessing gene expression regulation in female reproductive trait tissues are still limited, and this information is crucial to correctly interpret GWAS signals.

Considering the reviewer's comment and aiming for data presentation clarity, we have refined our approach for our gene prioritization criteria. The selected genes are now considered regardless of regulatory data availability (for instance nearest gene or relevant mouse model phenotype were accounted regardless of regulatory data availability of a certain gene).

We believe that the inclusion of different layers of evidence (positional, coding, regulatory and altering phenotypes in mice models), offers a balanced overview of plausible candidate genes in each locus considering that some relevant tissue for this gynecological condition might not be available or underrepresented with small sample sizes.

Line 116-124 (Results) and 401 (Methods):

"In order to move from genetic variants to plausible candidate genes, we prioritized genes according to different data layers of evidence, considering at least the presence from one of the next four main evidence levels: 1) positional mapping as implemented in FUMA v1.6.3 was used to determine the nearest gene to the association peak; 2) genes containing shared causal variants between genetic variants and gene expression signatures unraveled by colocalization analyses; 3) genes containing coding variants or in high LD ($r^2 > 0.6$) with these; 4) genes which showed embryo, growth/size/body, muscle, renal/urinary system, reproductive system, digestive/alimentary system phenotypes in mutant mice."

In this regard, we acknowledged and emphasized the need of further studies assessing gene expression in relevant tissues in discussion in line 384:

"Future larger sample size in gene expression panels of relevant tissues in gynecological conditions is crucial, since might add greater power to detect more significant associations and improve disentangling tissue specific signals in eQTL datasets"

3. Genetic correlations: For the traits identified with significant genetic correlations, which studies were these from and did they overlap with your meta-analysis sample?

The 90 significant traits presented in the text and past supplementary table 3 (now Supplementary Data 7), come mainly from UKB publicly available GWAS summary statistics from Benjamin Neale lab (<http://www.nealelab.is/uk-biobank/>) (N=86 studies). The additional four traits come respectively from anthropometric (Waist-to-hip ratio PMID: 25673412), cardiometabolic (Coronary artery disease, PMID: 26343387), lipids (Triglycerides, PMID: 20686565) and reproductive traits categories (Age of first birth, PMID: 27798627).

We thank the reviewer for raising the point of a possible sample overlap, which indeed it is the case since the presented meta-analyses contains data from UKB. The other studies,

assessing waist-to-hip ratio, coronary artery disease, lipids and age of first birth result from large consortium efforts which overlap as well with Estonian Biobank, UKB and FinnGen data. Nonetheless, the method utilized for estimating genetic correlation, cross-trait LD Score regression – requires only GWAS summary statistics and is not biased by sample overlap (<https://www.ncbi.nlm.nih.gov/pmc/articles/PMC4797329/>).

In order to support that sample overlap does not affect the presented analysis, we have conducted an additional analysis testing the genetic correlation between the Estonian Biobank and FinnGen fraction of the study (including 20,528 cases and 384,100 controls) against the 516 available traits from UKBB, thus avoiding sample overlap between studies. When comparing the most significant 5 genetic correlation estimates presented in Figure 3 of our manuscript (red dots) to the new non-overlapped genetic correlation analysis (blue dots), we observe that genetic correlations are similar between runs. Whilst observing an expected weaker signal in p-value, since EstBB+FinnGen sample size is smaller, the associations reached nominal significance ($p < 0.05$).

We have now mentioned that the method utilized is not biased by sample overlap in Methods and add the reference in line 430 (PMID: 26414676). Additionally, we have added in Supplementary Data 7 the results of genetic correlation analyses for all 561 traits tested and the source of respective studies.

4. Genetic correlations: Was the timing of hysterectomy in relation to POP considered – I think this can be a cause of but also in some cases a treatment for prolapse? It could be interesting to explore this further.

We acknowledge that a limitation of our study is the unavailability of timing of hysterectomy in relation to POP in genetic correlation analyses, which would allow us a better understanding of the effect hysterectomy might have as both cause and consequence for prolapse.

Genetic correlation between pelvic organ prolapse and Ever had hysterectomy was conducted utilizing publicly available summary statistics for the trait ‘Ever had hysterectomy (womb removed)’ from Benjamin Neale lab, where 171413 women answered to the question

'Have you had a hysterectomy (womb removed)? From those, 13973 answered 'yes' (cases) and 157440 answered 'no' (controls). Therefore, further dissection of the timing of hysterectomy in relation to POP is unfortunately unavailable in this analysis, since we do not have access to individual level data.

In respect to the other studies included in this study, similarly we do not have access to individual level data from Icelandic and UKB and FinnGen study, where summary level data was utilised. In Estonian Biobank, operation codes are available for 42.9% women with POP (3,563 out of 7,968). POP-related procedure codes (corresponding to vaginal plastic surgery, vaginal reconstruction with its own tissues, vaginal reconstruction with transplant, vaginal resection, laparoscopic saccholepopexy or lateral suspension, vaginal hysterectomy, radical hysterectomy, radical hysterectomy type B, laparoscopic hysterectomy and laparoscopic-assisted vaginal hysterectomy) are present in 11.4% of the cases (909) and only 1% (95) present POP-related surgery before getting N81 diagnose. Similarly, 112 women present hysterectomy-related procedure codes (corresponding to vaginal hysterectomy, radical hysterectomy, radical hysterectomy type B, laparoscopic hysterectomy and laparoscopic-assisted vaginal hysterectomy) and only 15 present an earlier date of hysterectomy than prolapse diagnosis. We believe the sample size is underpowered to draw conclusions about these relationships. Nevertheless, future linkages between the Estonian Biobank and the Estonian National Health Insurance Fund Registry (a central digital registry which collects data on procedure codes in Estonia) will bring the opportunity to assess the relationship between hysterectomy and pelvic organ prolapse and we are looking forward to having the possibility to assess this relationship in the near future.

We have acknowledged the interest to follow-up this aspect in discussion.

Discussion, line 306-309: "While unfortunately it is not possible to discern the timing of hysterectomy in relation to POP in the present study, future linkages and/or cohorts with larger availability of POP-related procedures might better inform the effect of hysterectomy as both cause and consequence to POP."

5. Prediction: Was age included as a variable in all prediction models and what effect does this have on the C-statistics over PRS?

Testing the predictive ability of the PRS was done in the validation set (2,517 incident cases and 96,109 controls). The validation subset (2,104 incident cases and 24,780 controls) where clinical variables were also taken into account alone or in combination with the PRS. In both sets, we used survival modeling to conduct the PRS prediction ability analyses, which were properly accounted for age by including it as the time scale in cox proportional hazard models to properly account for left-truncation and right-censoring in the data. In this way, we compared only women from same ages and avoided inflation in prediction solely due to age difference between cases and controls.

We hope we reinforce this point in lines adding the following statement in Discussion, line 320-323: It is important to note that predictive ability analyses in the validation set properly took into account the effect of age by including it as the time scale in the survival model, thus properly accounting for left-truncation and right-censoring in the data, and comparing only women from same ages and avoiding inflation in prediction solely due to age difference between cases and controls.

Methods, line 489 "We used survival modeling, where age was used as a timescale to properly account for left-truncation in the data and right-censoring."

Supplementary Data 11. Summary statistics between cases and controls through different groups defined are given.

Dataset/analysis	GWAS/association analysis		Discovery set/PRS best version selection		Validation set/PRS predictive ability		Validation subset/PRS and clinical risk factors predictive ability	
	Cases	Controls	Prevalent cases	Controls	Incident cases	Controls	Incident cases	Controls
Sample size (N)	7896	118695	5379	21516	2517	96109	2104	24753
Age at agreement (average, (standard deviation))	58.76 (12.01)	43.86 (16.06)	61.03 (11.22)	43.8 (15.92)	53.92 (12.20)	43.88 (16.07)	53.26 (12.21)	44.45 (17.24)
<40 years old (N)	519	51533	204	9482	315	42051	281	10829
[40-50) years old (N)	1195	24164	640	4424	555	19740	483	4577
[50-60) years old (N)	2271	19987	1437	3621	834	16366	701	4058
60-70 years old (N)	2421	13473	1854	2455	567	11018	457	2856
70-80 years old (N)	1251	6740	1031	1241	220	5499	166	1832
80-90 years old (N)	222	1595	200	269	22	1326	13	552
>90 years old	16	129	13	24	3	108	3	49
Was the model adjusted by age?	Yes		Yes, age and age squared		Cox proportional hazard models were applied, and age was used as the time scale, comparing same age groups			

6. Prediction: How were the clinical risk factors chosen for the prediction model?

Clinical risk factors included in the predictive model were chosen based on three main aspects:

1) *likely involvement in pelvic organ prolapse identified through literature mining, including previously described risk factors (PMID: 25966804/, PMID: 27439423 , PMID: 15902178 ,PMID: 19444367/). Whilst some evidence linked POP severity and bronchial asthma(https://erj.ersjournals.com/content/38/Suppl_55/p521), our study concludes asthma is a poor predictor, in line with recent research which suggested that asthma is not a risk factor for POP(PMID:33873084).*

2) *in discussion with clinicians who are treating pelvic organ prolapse and are among the authors of this paper*

3) *this selection was also influenced by the availability of clinical risk data in Estonian Biobank data.*

which we now referred in Methods, line 494:

“Clinical risk factors were chosen according to literature mining and data availability in Estonian Biobank.”

Discussion, line 354-357

“Similarly, the present study is limited by the unavailability of other classical risk factors that would improve predictive ability of the combined model, such as newborn anthropometric measurements, mode of delivery, or more extensive reproductive history.”

7. Discussion: I think the following statement should be more cautious: “this is the first study evaluating a PRS for a female-specific non-cancer trait” - there are other studies employing similar methods e.g. Rafnar et al. Nat Comms 2018 doi:10.1038/s41467-018-05428-6; Ruth et al. Nature 596 2021 doi: 10.1038/s41586-021-03779-7; Joo YY et al. J Clin Endocrinol Metab. 2020 doi: 10.1210/clinem/dgz326.

We agree with the reviewer and have revised this sentence as suggested in the following lines:

Line 318: “The added value of our study in comparison to previous efforts¹² is best highlighted by presenting the first polygenic risk score in POP, which substantially advances the concept of using genomic information to stratify women in gynecological conditions.”

Line 369. “In summary, this study is the first to explore polygenic risk scores as a tool to inform risk stratification strategies in POP and highlights the potential to improve predictive ability when adding genetic risk on top of clinical risk factors“.

8. Methods: Some further detail is required to describe how the independent significant loci were identified and any MAF and imputation threshold applied to the GWAS results.

We agree with the reviewer and had discussed that it would be needed to add further detail about how independent significant loci were identified and MAF/imputation thresholds applied to the GWAS results. Below we detailed how these were applied for each study:

We have added these details in the following sections and lines:

In Supplementary Methods:

Line 4-6: Summary statistics were downloaded upon request, which included prefiltered variants selected based on a threshold of 0.8 imputation info and MAF > 0.01%, available in the Icelandic data set and/or the UKB dataset.

Line 14-15: “which included prefiltered variants (minimum allele count >5 and INFO score >0.6).”

Line 45-46: “In EstBB, SNVs with poor imputation quality (INFO score<0.4) and minor allele count <5 were excluded from downstream association analysis.”

In Methods:

Line 389-396 of Methods: “A total of 40,542,692 variants were included in the meta-analysis of 28,086 women with POP and 546,291 female controls. After meta-analysis we kept variants present in minimum 2 out of the 3 studies (totaling 15,871,689 variants) for downstream analysis. Details of filters applied in pre-association analyses stage of the three studies meta-analysed are provided in Supplementary Methods. Lead SNPs were identified as SNVs independent from each other with P-value less than or equal 5×10^{-8} . 500kb was set to the maximum distance between LD blocks of independent significant SNPs to merge into a single genomic locus.”

The inclusion of chromosome X from Estonian Biobank slightly affect the p-values of gene-set and tissue/cell type specific enrichment, which are now corrected in Results and Supplementary Figure 3.

Line 160-167, Results:

Gene set and tissue/cell-type enrichment. Gene set and tissue/cell-type enrichment analysis implemented in MAGMA¹⁶ and DEPICT¹⁷ highlighted “Connective Tissue Development” ($p=2.01 \times 10^{-6}$), “Chondrocyte differentiation” ($p=6.63 \times 10^{-4}$), “In utero embryonic development” ($p=4.91 \times 10^{-8}$), “Abnormal embryonic tissue morphology” ($p=9.43 \times 10^{-7}$) and “Small heart” ($p=9.8 \times 10^{-6}$), Supplementary Figure 3 and Supplementary Data 3 and 4). 12 tissues were significantly enriched after correcting for multiple testing, including “Cervix/ectocervix” ($p=1.30 \times 10^{-5}$), “Uterus” ($p=1.50 \times 10^{-5}$), “Embryoid bodies” ($p=8.60 \times 10^{-6}$) and “Smooth muscle” ($p=7.30 \times 10^{-4}$; Supplementary Figure 3 and Supplementary Table Data 5 and 6).

9. Methods: Please clarify the meaning of “poor effect first trimester miscarriages likely have in prolapse”.

It is well known that parity is a strong risk factor for pelvic organ prolapse (PMID: 25966804/). However, the weakening of the pelvic floor happens during later stages of pregnancy (by extra weight and inherent weakness of pelvic support structures (PMID: 29332252/, 26875952/, PMID: 19095964/) and hormonal changes which might affect qualitative properties of pelvic floor supportive tissues (PMID: 18951211/). Thus, an increased risk of pelvic organ prolapse due to first trimester miscarriages seems to be unlikely and most likely the effect of pregnancy in pelvic organ prolapse risk is better captured in second and third trimesters of pregnancy.

We have rephrased this statement in line 500-502, which we hope provides a clearer information now: “excluding possible first trimester miscarriages (which are not affected by extra weight and weakness of pelvic floor structure and thus are unlikely to contribute to prolapse development).”

10. Supplementary Figure 3 and 4 should be Supplementary Figure 4 and 5, etc.

We have now fixed the error on the annotation of Supplementary Figures. We have corrected these to their correct forms throughout the text and figure legends.

REVIEWERS' COMMENTS

Reviewer #1 (Remarks to the Author):

My comments have been adequately addressed.

Reviewer #2 (Remarks to the Author):

The authors have comprehensively answered my comments and made appropriate changes that strengthen this interesting paper. Two very minor comments remain:

Table 1. rs28403275 – EA has been auto-formatted.

Supplementary Methods line 73: Subheading with no following text – details of PRS calculation not present.

Reviewer #3 (Remarks to the Author):

The revised manuscript "Advancing our understanding of genetic risk factors and potential personalized strategies in pelvic organ prolapse" by Pujol-Gualdo et al. represents an advance in genetics of POP compared to previous GWAS. In particular, the new study, which is done in a thoughtful and careful way, increases the number of independent loci from 7 to 30, highlights relevant biological pathways that are consistent with the clinical features of POP, and begins to explore the potential translation of genetic susceptibility to POP to improve health. Here are comments/suggestions, all relatively minor, requesting clarifications to the text and possibly very modest additional analysis.

Abstract

While PRSs have been getting attention recently, the claims about potential translation are often exaggerated, and certainly rarely vetted in the clinical setting. The current analysis is better than many in looking at genetic effects in the prospective setting, which is preferred. Moreover, the genetic effects are relatively larger than for some other outcomes. Nevertheless, the total SNP based effect on the liability scale is only 9.4%. This is the maximum possible for common genetic variation in the authors' approach. This is a subjective point, but the final conclusion in the abstract about personalized risk prediction may be somewhat exaggerated. See also comments below.

Introduction

Line 66. Recommend giving some context about how much larger the current study is than previous studies, e.g. how many times more cases are there in this work than in previous analysis?

Results

Line 97: Referring to heterogeneity of the loci, the authors might specify that this sentence addresses to all 30 signals at once, e.g. "... no heterogeneity across the three data sets at any of the lead signals ...". They might also provide the range of I² values as %, noting that none was significant, e.g. an extra column in Table 1. This may be important, in part, because the utility of the Q-stat p-value is sometimes viewed as varying with number of studies included, e.g. too liberal with few studies and too severe with many studies.

Line 102. The authors run LDSC for a liability scale heritability estimate. LDSC would have also given evidence for potential inflation of the summary statistics via the intercept, as a complement to the genomic control value that the present. Would the authors include the LDSC intercept stats? The significance of this term is the important thing.

Line 103. The SNP-based liability is estimated as 9.4%. How much of this is due to the 30 independent lead SNPs alone?

Line 120. "genes containing shared causal variants between genetic variants" is a somewhat confusing statement. Would the authors clarify?

Line 122. "genes containing coding variants or in high LD ..." Do the authors mean "non-synonymous" or "coding" as written?

Line 210. Please specify the scale of the association. Is this an OR per SD of the PRS or something else?

Lines 233-235. Is the improvement in the AUC by adding the PRS significant? The authors could get an empirical p-value by resampling the PRS without replacement to get a reference null distribution of the C-statistic for the models with the clinical covariates and a null PRS. Then they could compare the observed improvement to this empirical null distribution.

PRS analysis in general. In the response to reviewers' document, the authors explain how they use age as the time variable in the Cox models of the PRS to deal with competing risks. A related issue is whether the authors can test for an interaction of the POP PRS with age. This analysis would address the issue of whether the PRS may be relatively more useful at older or younger ages. Is this something that can be examined in the study sample?

Discussion

Line 322. Despite the value of the PRS analysis, it would be responsible to place limitations on the maximum potential for genetics as a predictor. The potential is roughly limited by heritability due to common variation, here estimated as about 9.4% on the liability scale under the additive model. See also comment above.

Line 341. Potential application of the PRS. The authors suggest some scenarios for potential use of the PRS, mostly in elevating awareness of risk. A critical issue is whether POP is a reversible condition, e.g. by weight loss and/or exercise. Can the authors comment on whether the PRS may be particularly relevant to treatment, e.g. especially helpful for anticipating cases of POP that are irreversible? Are there any circumstances where genetic risk assessment might help for diagnosis, e.g. choosing between alternative possible diagnoses?

Line 350. The authors address phenotype definition here, highlighting the EstBB. Can they also add a few thoughts about misclassification, either unreported POP in the controls or incorrect diagnoses/diagnostic heterogeneity in the cases? Are there prior studies in the epidemiology literature that have estimated the limits of misclassification in the types of populations and ascertainment approaches that were applied in this study? More detail on this may be particularly relevant to the discovery and validation sets for the PRS analysis.

It's not critical, but the authors could talk about the discrepancy between the estimated 43% (in the Intro) of risk attributed to genetics and the observed 9.4% of the liability explained by common SNPs.

The Discussion could address relationships between the genetic correlations they report and

epidemiological, i.e. phenotypic, correlations, e.g. in the UKBB or elsewhere.?

Methods

Line 407. Is the word "Then" not needed?

Line 460. Please motivate the 19 versions of the PRS, e.g. the methods show 2 methods x 8 p-value thresholds = 16. What are the 3 additional versions? Are all PRSs standardized to mean=0, SD=1. Regardless, the authors should include units (e.g. SDs) in reporting ORs for the PRS associations.

Section beginning 467. The language about the incident POP data set is a little ambiguous. Cases seem to be individuals who had not been diagnosed with POP when surveillance began but who developed POP while under observation. It's not clear how controls were ascertained. Are these individuals otherwise were ascertained on the basis of similar criteria to the cases when the surveillance began? Please confirm and describe/clarify as needed.

REVIEWERS' COMMENTS

Reviewer #1 (Remarks to the Author):

My comments have been adequately addressed.

Reviewer #2 (Remarks to the Author):

The authors have comprehensively answered my comments and made appropriate changes that strengthen this interesting paper. Two very minor comments remain:

Table 1. rs28403275 – EA has been auto-formatted.

We have corrected this auto-formatting mistake in Table 1, now placed at the end of the manuscript.

Supplementary Methods line 73: Subheading with no following text – details of PRS calculation not present.

Following editorial requests we have now merged Methods and Supplementary Methods into a unique section 'Methods' and we are presenting details of PRS predictive ability analyses below line 564 of Methods.

Predictive ability of the PRS

We standardised the best PRS version and also categorized it into different percentiles (<5%, 5%-15%, 15%-25%, 25%-50%, 50%-75%, 75%-85%, 85%-95%, >95%). Survival modeling and Cox proportional hazard models were used to estimate the Hazard Ratios (HR) corresponding to one standard deviation (SD) of the continuous PRS for the validation set. Harrell's C-statistic was used to characterize the discriminative ability of each PRS. Cumulative incidence estimates were computed using the Kaplan-Meier method. We used survival modeling, where age was used as a timescale to properly account for left-truncation in the data and right-censoring. Cox proportional hazard models were also used as specified before to assess the differences between genetic risk in different age strata (women <40 years old (y), 40-50y, 50-60, 60-70y, >70y).

Reviewer #3 (Remarks to the Author):

The revised manuscript “Advancing our understanding of genetic risk factors and potential personalized strategies in pelvic organ prolapse” by Pujol-Gualdo et al. represents an advance in genetics of POP compared to previous GWAS. In particular, the new study, which is done in a thoughtful and careful way, increases the number of independent loci from 7 to 30, highlights relevant biological pathways that are consistent with the clinical features of POP, and begins to explore the potential translation of genetic susceptibility to POP to improve health. Here are comments/suggestions, all relatively minor, requesting clarifications to the text and possibly very modest additional analysis.

Abstract

While PRSs have been getting attention recently, the claims about potential translation are often exaggerated, and certainly rarely vetted in the clinical setting. The current analysis is better than many in looking at genetic effects in the prospective setting, which is preferred. Moreover, the genetic effects are relatively larger than for some other outcomes. Nevertheless, the total SNP based effect on the liability scale is only 9.4%. This is the maximum possible for common genetic variation in the authors’ approach. This is a subjective point, but the final conclusion in the abstract about personalized risk prediction may be somewhat exaggerated. See also comments below.

We have toned down a bit our latest statement in the abstract as follows:

“These findings improve our understanding of genetic factors underlying POP and provide a solid start evaluating PRS as a potential tool to enhance individual risk prediction.”

Introduction

Line 66. Recommend giving some context about how much larger the current study is than previous studies, e.g. how many times more cases are there in this work than in previous analysis?

We have added the estimated gain in number of cases compared to previous efforts in line 67:

In this work, we present the largest GWAS in POP to date (nearly doubling the amount of cases compared to previous efforts (PMID: 32184442) and systematically dissect the association signals to propose potential causal genes in associated loci.

Results

Line 97: Referring to heterogeneity of the loci, the authors might specify that this sentence addresses to all 30 signals at once, e.g. “... no heterogeneity across the three data sets at any of the lead signals ...”. They might also provide the range of I² values as %, noting that none was significant, e.g. an extra column in Table 1. This may be important, in part, because the utility of the Q-stat p-value is sometimes viewed as varying with number of studies included, e.g. too liberal with few studies and too severe with many studies.

In the revised version of the manuscript we have added I² values as % for all lead signals (see extra column in Table 1, now at the end of the manuscript). We observe that according to I², there are two signals with large heterogeneity (I²>75%) (rs72624976, I²=83.46% and rs3820282 I²= 76,45%).

We have added the following statements in Results, line 97:

According to I², two variants showed large heterogeneity (rs3820282 I²=76,45% and rs72624976, I²=83.46%, the latter likely due to differences in allele frequency between cohorts (Supplementary Table 1), although Q-Cochran test showed no heterogeneity of effects

between the three datasets at any of the lead signals (Q-Cochran p varying from 0.0023-0.982, with a Bonferroni corrected threshold of $0.05/30=0.001$) (Supplementary Figure 2, Table 1).

We have stated the heterogeneity tests in Methods line 449:

We performed two tests of heterogeneity of effects across studies (Q-Cochran test and heterogeneity index calculation (I^2)) as implemented in GWAMA (v2.2.2).

and discuss potential explanations for those in Discussion, line 331:

We observed that heterogeneity estimates of lead signals (taking into account Q-Cochran p-values and I^2) mostly do not substantially differ between studies, which on one hand adds reliability to the identified genetic associations and on the other hand supports leveraging electronic health records to ascertain this phenotype. According to I^2 , two variants showed heterogeneity (rs3820282 $I^2=76.45\%$ and rs72624976, $I^2=83.46\%$). rs72624976 heterogeneity is likely due to differences in allele frequency between cohorts (see Supplementary Table 1 and Supplementary Figure 1, with lower frequency for FinnGen and EstBB (EAF=0.01) than Icelandic and UKBB (EAF=0.04)). This observation emphasizes the importance of the existence of population-specific biobanks when studying genetics of complex diseases. rs3820202 heterogeneity might be explained by the presence of more severe cases in FinnGen and IceUK cohorts (perhaps due to age distribution differences between cohorts or source of diagnoses: hospital inpatient registry data for UKBB/FinnGen which might capture more severe cases compared to EstBB, where diagnoses made by general practitioners or gynecologists can also include milder diagnoses). However, heterogeneity estimates tend to be imprecise when assessing heterogeneity in meta-analysis containing a small number of studies (PMID: 25880989, PMID: 16784338, PMID: 17786212).

Line 102. The authors run LDSC for a liability scale heritability estimate. LDSC would have also given evidence for potential inflation of the summary statistics via the

intercept, as a complement to the genomic control value that the present. Would the authors include the LDSC intercept stats? The significance of this term is the important thing.

We have now included the LDSC intercept stats (LDSC intercept= 1.0059 (s.e. 0.0079) when pointing out the SNP-heritability calculated in LDSC in line 108.

Line 103. The SNP-based liability is estimated as 9.4%. How much of this is due to the 30 independent lead SNPs alone?

We observed the significant loci identified in this study capture 2.2% of SNP-heritability for pelvic organ prolapse, a small part of pelvic organ prolapse heritability.

We added in line 109: The SNP heritability explained by the significant loci identified was 0.02 given a population prevalence of 0.05.

We mentioned the additional analysis in Methods, line 453: After excluding genome-wide significant loci (variants within +/- 500kb from lead signals), we calculated the SNP heritability of the remaining variants and the one corresponding to the significant loci.

Line 120. “genes containing shared causal variants between genetic variants” is a somewhat confusing statement. Would the authors clarify?

We hope to have made a clearer formulation of this statement as follows, in line 117: “genes containing variants which showed significant (posterior probability >0.8) colocalization in eQTL datasets”.

Line 122. “genes containing coding variants or in high LD ...” Do the authors mean “non-synonymous” or “coding” as written?

We have now clarified this statement refers to non-synonymous variants and corrected in the text, line 118: “genes containing non-synonymous variants”.

Line 210. Please specify the scale of the association. Is this an OR per SD of the PRS or something else?

We have now clarified the scale of the association in line 182:

“The resulting PRSs associations were scaled as follows: we calculated odds ratio (OR) per one standard deviation of the PRS and 95% confidence intervals (95% CI).”

Lines 233-235. Is the improvement in the AUC by adding the PRS significant? The authors could get an empirical p-value by resampling the PRS without replacement to get a reference null distribution of the C-statistic for the models with the clinical covariates and a null PRS. Then they could compare the observed improvement to this empirical null distribution.

We thank the reviewer for this interesting comment. We have obtained an empirical p-value after making 10,000 random iterations of the PRS distribution in the validation subset. We then compared the proportion of times the PRS presented in the study showed a higher C-statistic by comparing to the mean C-stat of these 10,000 random PRSs (in the cox proportional hazard model adjusted by clinical variables). This proportion is 0.9999, which results in an empirical p-value of $1 - \text{proportion} = 9.999e-05$, which supports that the PRS evaluated in the present study substantially adds in the predictive ability of incident POP, above five established clinical risk factors.

In the histogram below we show the distribution of C-stat when adding PRS random in the model adjusted by clinical variables (blue), and we observe that in all cases this value is lower than the C-stat obtained when adding real PRS + clinical variables (C-stat=0.629) (green dashed line) and in most cases than the model when only adjusting for the five clinical variables alone (red dashed line).

PRS analysis in general. In the response to reviewers' document, the authors explain how they use age as the time variable in the Cox models of the PRS to deal with competing risks. A related issue is whether the authors can test for an interaction of the POP PRS with age. This analysis would address the issue of whether the PRS may be relatively more useful at older or younger ages. Is this something that can be examined in the study sample?

We thank the reviewer for this suggestion. We have now stratified the discovery set into different age categories; <40 years old (y), 40-50 y, 50-60 y, 60-70y and >70y and obtained HR and 95% CI using survival modeling. We have calculated HR for 1) continuous distribution of the PRS 2) top 5% genetic risk versus rest 3) top 5% genetic risk versus average genetic risk. Results are shown in the table at the end of this comment, which we have added as a new supplementary table – **Supplementary Data 12.**

Although we observe stronger association between incident status and genetic risk in those younger age strata compared to older strata, highest HR is observed in the 50-60 year-old group, which also has the largest case sample size in incident analysis (n cases= 857).

Overall, we do not see any age strata that shows an increased association compared to the full validation set and the assessment of genetic risk effects in the genetic risk categories in older age stratum (>70y) is hindered by very low cases sample size in in top 5% genetic risk ($n=8$).

In addition, we want to clarify that in our study we have not accounted for competing risks. We have accounted for left censoring to adjust for the fact that some individuals cannot join the biobank due to health problems or due to death before joining the biobank and right censoring to adjust for the fact that we have limited follow-up period. However, competing risks (i.e. POP vs death or illness type of models were POP, death or POP to death transitions would be considered) are not currently investigated here. We do acknowledge that this might be a limitation of our study, however we expect the effect of death to be relatively small on pop incidence in early and middle-life ages.

We have added a brief comment on this extra analysis in the manuscript:

Methods, line 572: Cox proportional hazard models were also used as specified before to assess the differences between genetic risk in different age strata (women <40 years old (y), 40-50y, 50-60, 60-70y, >70y).

Results, line 198: When assessing genetic risk in different age categories, none of the age groups showed a higher HR compared to the full validation set analysis (Supplementary Data 12). Women in 50-60 years old strata showed highest HR amongst categories when comparing top 5% genetic risk vs rest of women (HR=2.04, 95% CI=1.42-2.70 $p=6.6 \times 10^{-7}$), as well as highest number of incident cases in top 5% genetic risk ($n=834$) compared to younger and older strata.

	PRS as continuous			Top 5% genetic risk vs rest of women			Top 5% of PRS vs average			Sample size			
	Hazard Ratio	95% CI	P value	Hazard Ratio	95% CI	P value	Hazard Ratio	95% CI	P value	N incident cases	N controls	N of cases in top 5% genetic risk	Total N in top 5% genetic risk
Validation set	1,31	1,25-1,37	<2e-16	1,61	1,35-1,92	6.64e-08	1,53	1,26-1,86	1.7e-05	2517	96109	183	4722
[30-40) years old	1,33	1,16-1,54	5.23e-05	1,45	0,84-2,49	0,17	1,46	0,79-2,69	0,22	239	20340	17	1006
[40-50) years old	1,24	1,13-1,36	5.02e-06	1,55	1,06-2,26	0,02	1,42	0,94-2,16	0,09	555	19710	39	963
[50-60) years old	1,33	1,16-1,54	5.23e-05	2,04	1,54-2,70	6,60E-07	1,96	1,42-2,70	4,09E-05	834	16239	73	827
[60-70) years old	1,26	1,14-1,38	1.55e-06	1,44	0,99-2,08	0,05	1,48	0,97-2,25	0,06	567	11099	38	541
>=70 years old	1,32	1,12-1,55	0,0008	0,61	0,22-1,66	0,315	0,47	0,16-1,32	0,156	245	6992	8	311

Discussion

Line 322. Despite the value of the PRS analysis, it would be responsible to place limitations on the maximum potential for genetics as a predictor. The potential is roughly limited by heritability due to common variation, here estimated as about 9.4% on the liability scale under the additive model. See also comment above.

We have added a comment considering the limitation of PRS as a predictor given the still low heritability that common variation explains for this condition.

Line 294: However, it is prudent to consider that common SNPs explain a small part of the whole heritability and it is plausible that most of SNP-heritability is yet to be discovered, which hinders assessing the full potential of PRS. Potential sources of missing heritability might include much larger numbers of smaller effect variants yet to be found; rarer variants (possibly with larger effects) that are poorly detected by available genotyping arrays that focus on variants present in 5% or more of the population; structural variants poorly captured by existing arrays; low power to detect gene-gene interactions, etc (PMID: 19812666). Further studies with larger sample sizes are needed, which will enable comprehensive PRS performance comparisons and will improve the evaluation of genetic risk assessment.

Line 341. Potential application of the PRS. The authors suggest some scenarios for potential use of the PRS, mostly in elevating awareness of risk. A critical issue is whether POP is a reversible condition, e.g. by weight loss and/or exercise. Can the authors comment on whether the PRS may be particularly relevant to treatment, e.g. especially helpful for anticipating cases of POP that are irreversible? Are there any circumstances where genetic risk assessment might help for diagnosis, e.g. choosing between alternative possible diagnoses?

We thank the reviewer for raising this point. We have reinforced in the text that one clinical utility of genetic risk as a predictor would be to anticipate those severe cases – although remains to be assessed how much those would benefit from early lifestyle interventions. We added an additional possibility of PRS application in future cohorts when stratifying cases severity in line 326: **Similarly, future studies assessing risk prediction towards those cases who present more severe forms - e.g. requiring surgical intervention- might open up new avenues for targeting clinical resources, increasing check-up frequency and direct targeted preventive exercises and counseling for those women.**

We believe that the utility of PRS towards alternative diagnoses is rather limited since symptomatic POP can be well identified in gynecological/physical examinations.

Line 350. The authors address phenotype definition here, highlighting the EstBB. Can they also add a few thoughts about misclassification, either unreported POP in the controls or incorrect diagnoses/diagnostic heterogeneity in the cases? Are there prior studies in the epidemiology literature that have estimated the limits of misclassification in the types of populations and ascertainment approaches that were applied in this study? More detail on this may be particularly relevant to the discovery and validation sets for the PRS analysis.

We have commented around ascertainment of cases and controls in the biobank setting in line 350. Although misclassification is a common issue, our study prevalence suggests we have captured well those cases with symptomatology. Apart from Estonian Biobank, there are other studies supporting case ascertainment using ICD coding in Finnish Care Register for Health Care (PMID: 22899561, also see: <https://doi.org/10.5324/nje.v14i1.284>) and in the study performing first GWAS about POP (PMID: 32184442), we have mentioned and further discussed those aspects in Discussion, line 412:

In this regard, one of the strengths of the study is the comprehensive data availability in EstBB, containing genetic data of around 20% of Estonian adult population including phenotype

questionnaire and measurement panel, together with follow-up data from linkage with national health-related registries, which facilitated the validation of PRS and the inclusion of clinical risk factors into a joint model. In a similar way, the coverage and accuracy of the Finnish Care Register for Health Care has been validated previously (PMID: [22899561](https://pubmed.ncbi.nlm.nih.gov/22899561/), also see: <https://doi.org/10.5324/nje.v14i1.284>) and it has been found to be excellent. Previous effort meta-analyzing UKB and Icelandic studies assessed the robustness of ICD codes by comparing effect sizes with surgically treated POP cases and concluded that effects were in the same direction and not substantially different from those using surgically treated POP cases (PMID: [3218444](https://pubmed.ncbi.nlm.nih.gov/3218444/)).

Possible phenotypic misclassification (undiagnosed cases amongst controls) could result in heterogeneity in the analysis and interpretation of GWAS findings, reducing both the statistical power as well as the maximum number of significant associations observed , their effects magnitude and direction.

We hope that future efforts replicate our findings, either independent or stemming from increased numbers of cases in new data linkages in the included datasets, and more extensive questionnaire data collection. Similarly, future questionnaires in the biobank setting could address different disease stages and severity.

It's not critical, but the authors could talk about the discrepancy between the estimated 43% (in the Intro) of risk attributed to genetics and the observed 9.4% of the liability explained by common SNPs.

We have added this part in discussion, line 297: Potential sources of missing heritability might include much larger numbers of smaller effect variants yet to be found; rarer variants (possibly with larger effects) that are poorly detected by available genotyping arrays; structural variants poorly captured by existing arrays; low power to detect gene-gene interactions, etc. (PMID: [19812666](https://pubmed.ncbi.nlm.nih.gov/19812666/)).

The Discussion could address relationships between the genetic correlations they report and epidemiological, i.e. phenotypic, correlations, e.g. in the UKBB or elsewhere.

We have added a few comments about main epidemiological associations presented to date in POP and how these reflect genetic correlation presented in this study in line 333:

Genetic correlation studies mirror well the findings of epidemiological studies, showing associations with number of births, previous hysterectomy, younger age at first birth, increasing BMI, constipation, occupations including heavy lifting and connective tissue disorders (PMID: 9166201, 25111588).

The associations between POP and gastroesophageal reflux, diverticular disease, osteoarthritis and hiatus hernia most likely reflect the changes to connective tissue characteristic to all of these conditions; however, respective epidemiological links have been inconsistent. Similarly, future epidemiological studies assessing the association with abdominal and pelvic pain, excessive frequent and irregular menstruation and cardiometabolic phenotypes are warranted.

Methods

Line 407. Is the word “Then” not needed?

We have removed ‘Then’ from the sentence in line 519:

We prioritised candidate genes considering four main evidence levels

Line 460. Please motivate the 19 versions of the PRS, e.g. the methods show 2 methods x 8 p-value thresholds = 16. What are the 3 additional versions? Are all PRSs standardized to mean=0, SD=1. Regardless, the authors should include units (e.g. SDs) in reporting ORs for the PRS associations.

In our study we used 2 methods, one uses p-value thresholds to include SNPs in the PRS (PRSice2) and second, includes all the SNPs in all PRS scores but modifies individual SNP weights under different assumptions of how many of these SNPs are causal (resulting in a total of 10 models). We have clarified the number of models analyzed in line 588 of Methods, as detailed below. We believe that the selection of different thresholds in the case of PRSice-2 is well justified since it adds a decreasing number of SNPs (from highest number $p \leq 1$, to lowest number $p < 0.0001$). The 19 versions of the PRSs are defined and we have now added the number of SNPs/model in Supplementary Data 10 (All scores are standardized to be with mean 0 and SD 1, so all ORs are per 1 SD in PRS associations). We now have added S.D. in the text and in Figure 6a and added the following information in Methods, line 516.

PRSice-2 uses a “clumping and thresholding” approach to clump genetic variants in close linkage disequilibrium²⁶, such that the remaining variants are independent of each other, and includes only those variants with a GWAS association *P*-value below a given threshold, with the threshold chosen to maximize the association of the risk score with POP. We tested the following 9 thresholds: 1, 0.3, 0.1, 0.03, 0.01, 0.003, 0.001, 0.0003 and 0.0001, with a maximum LD between them set to $r^2=0.2$. The number of SNPs included in each model are presented in Supplementary Data 10.

LDpred is a Bayesian approach that applies a continuous shrinkage model to modify effect sizes of SNPs to incorporate information on the strength of each variant’s association in the GWAS and the underlying linkage disequilibrium structure (PMID: 26430803). To decrease the dimension of multicollinearity, SNPs were clumped with maximum LD between them set to $r^2=0.99$. Then, 10 versions of PRSs were calculated by varying the fraction of causal SNPs on these values: Inf, 1, 0.3, 0.1, 0.03, 0.01, 0.003, 0.001, 0.0003 and 0.0001.

Section beginning 467. The language about the incident POP data set is a little ambiguous. Cases seem to be individuals who had not been diagnosed with POP when surveillance began but who developed POP while under observation. It’s not clear how controls were ascertained. Are these individuals otherwise were

**ascertained on the basis of similar criteria to the cases when the surveillance began?
Please confirm and describe/clarify as needed.**

We have added more details in the explanation and hope now is clearer, line 544:

The selection of controls in the discovery set was randomized, including 4 controls per case. Controls were defined as women who did not develop pelvic organ prolapse during follow-up (follow-up was initiated in first diagnoses linkage from Estonian Health Insurance Fund to Estonian Biobank in 26-11-2002 and ends in latest linkage to diagnoses dating from October 30-12-2019). Cases were not otherwise matched to controls.

We also added in line 424:

We excluded from all analyses controls who did not have health registry information linked, since those might have not had the opportunity to register diagnose status.